# The DREAM complex represses growth in response to DNA damage in *Arabidopsis*

Lucas Lang[1],*, Aladár Pettkó-Szandtner[2,3],*, Hasibe Tunçay Elbaşı[1] , Hirotomo Takatsuka[7], Yuji Nomoto[7], Ahmad Zaki[4,8], Stefan Dorokhov[4], Geert De Jaeger[5,6], Dominique Eeckhout[5,6], Masaki Ito[7], Zoltán Magyar[3], László Bögre[6], Maren Heese[1] , Arp Schnittger[1] 

**The DNA of all organisms is constantly damaged by physiological processes and environmental conditions. Upon persistent damage, plant growth and cell proliferation are reduced. Based on previous findings that RBR1, the only Arabidopsis homolog of the mammalian tumor suppressor gene retinoblastoma, plays a key role in the DNA damage response in plants, we unravel here the network of RBR1 interactors under DNA stress conditions. This led to the identification of homologs of every DREAM component in Arabidopsis, including previously not recognized homologs of LIN52. Interestingly, we also discovered NAC044, a mediator of DNA damage response in plants and close homolog of the major DNA damage regulator SOG1, to directly interact with RBR1 and the DREAM component LIN37B. Consistently, not only mutants in *NAC044* but also the double mutant of the two *LIN37* homologs and mutants for the DREAM component *E2FB* showed reduced sensitivities to DNA-damaging conditions. Our work indicates the existence of multiple DREAM complexes that work in conjunction with NAC044 to mediate growth arrest after DNA damage.**

## Introduction

The DNA of any organism is constantly damaged by intrinsic factors, such as reactive oxygen species and failures in DNA replication, as well as extrinsic conditions, for example, high energy radiation and the uptake of toxic compounds such as aluminum. Common cellular responses to DNA damage ranging from humans to plants include cell cycle arrest, transcriptional induction of DNA repair genes, and cell death to erase damaged cells. Whereas homologous genes between animals and plants are readily found at the level of the repair machinery itself, damage signaling and transcriptional regulation display fundamental differences between these kingdoms (Nisa et al, 2019).

A key DNA damage regulator in humans is the transcription factor p53, which is phosphorylated and thus stabilized after DNA damage, leading to the induction of many DNA repair genes and the repression of cell cycle-promoting genes. Whereas in many cases transcriptional activation by p53 is direct, down-regulation of cell cycle regulators is rather indirect. According to a current model (Hafner et al, 2019), repression is achieved by p53 promoting the transcription of p21, a CDK inhibitor, that leads to a reduction of CDK activity and thus, less phosphorylation of the retinoblastoma protein (pRb)-like proteins p107 and p130. Hypophosphorylated p107/p130 are then incorporated into the transcriptional repressor complex DREAM, composed of DP, pRb-like (p130 or p107), E2F, and the Multivulval class B (MuvB)-core, comprising five additional proteins, LIN9, LIN37, LIN52, LIN54, and RBBP4. In unperturbed cellular conditions, differential MuvB-core interactions are regulated in a cell cycle-specific manner, and the transcription-repressing DREAM complex is restricted to G0 and early G1 cells. In contrast to its homologs, the tumor suppressor pRb itself is not found as part of a DREAM complex but exerts its repressive function independently (Fischer & Müller, 2017). Thus, p53-mediated down-regulation of cell cycle genes after DNA damage can be separated into the repression of G1/S genes by mainly pRb, with some contribution of p130 and p107, and the repression of G2/M genes by p130 and p107 (Schade et al, 2019). In addition, the atypical E2F and transcriptional repressor E2F7, which is under direct transcriptional control of p53, is thought to function in conjunction with pRb and DREAM to mediate repression of cell cycle-related genes (Carvajal et al, 2012).

Notably, p53 is not conserved in all eukaryotes and a p53 homolog has not been identified in plants. Instead, the transcriptional regulator SOG1 is central to signal transduction after DNA damage (Yoshiyama et al, 2009; Yoshiyama, 2016). Upon DNA stress, SOG1 is

[1]Department of Developmental Biology, University of Hamburg, Institute for Plant Sciences and Microbiology, Hamburg, Germany   [2]Laboratory of Proteomic Research, Biological Research Centre, Szeged, Hungary   [3]Institute of Plant Biology, Biological Research Centre, Szeged, Hungary   [4]Department of Biological Sciences, Centre for Systems and Synthetic Biology, Royal Holloway University of London, Egham, UK   [5]Department of Plant Biotechnology and Bioinformatics, Ghent University, Ghent, Belgium   [6]Vlaams Instituut voor Biotechnologie (VIB) Center for Plant Systems Biology, Ghent, Belgium   [7]School of Biological Science and Technology, College of Science and Engineering, Kanazawa University, Kanazawa, Japan   [8]School of Life Sciences, University of Warwick, Coventry, UK

Correspondence: maren.heese@uni-hamburg.de; arp.schnittger@uni-hamburg.de
*Lucas Lang and Aladár Pettkó-Szandtner contributed equally to this work

phosphorylated and thus activated by the stress sensor kinases ATM/ATR (Yoshiyama et al, 2014). Similar to p53, SOG1 is upstream of a broad transcriptional program eventually leading to DNA repair, cell cycle arrest, cell differentiation, and/or cell death (Yoshiyama, 2016). SOG1 is, for example, a direct positive regulator of DNA repair genes as well as of the CDK inhibitors SMR5 and SMR7 and two closely related SOG1 homologs, NAC044 and NAC085 (Bourbousse et al, 2018). Although transcriptional targets of NAC044 and NAC085 are currently unknown, they have been hypothesized to indirectly control the stability of MYB3R3, a repressive transcription factor, shown to act on mitotic genes (Takahashi et al, 2019).

In contrast to p53, pRb-like proteins are greatly conserved between plants and animals. Previous work indicated that the only Arabidopsis pRb homolog RETINOBLASTOMA-RELATED 1 (RBR1) is another key regulator of the DNA damage response (DDR). On the one hand, RBR1 was found to accumulate in foci in the nucleus, to work together with the BREAST AND OVARIAN CANCER TYPE 1 SUSCEPTIBILITY PROTEIN (BRCA1) (Horvath et al, 2017), and to be necessary for the recruitment of the DNA repair machinery, such as RAD51, to DNA lesions (Biedermann et al, 2017). On the other hand, RBR1 was found to associate with promoters of many DDR genes and to regulate their expression in a SOG1-independent manner (Biedermann et al, 2017; Horvath et al, 2017; Bouyer et al, 2018), an interaction that likely represents a priming mechanism and couples DNA repair genes with cell proliferation.

Apart from its DNA damage targets, RBR1 has been shown to bind promoters of genes related to different cell cycle phases (Bouyer et al, 2018), and DREAM-like complexes have been identified to differentially regulate cell division activity also in plants (Kobayashi et al, 2015; Fischer & Müller, 2017). Interestingly, in this context, the existence of not only repressing but also activating DREAM complexes, depending on the type of MYB3R and E2F homologs involved, has been postulated. The complex activating mitotic genes was found to at least contain ALY3 (LIN9 homolog), TCX5 (LIN54 homolog), CDKA1, RBR1, MYB3R4, E2FB, and DPA/B, whereas for a presumptive repressive complex ALY2, ALY3, TCX5, RBR1, MYB3R3, and E2FC have been shown to interact. However, homologs of the MuvB-core components LIN37, LIN52, and RBBP4 have not been detected in this study (Kobayashi et al, 2015). Recently, TCX5/6-containing multi-subunit complexes were found to promote DNA demethylation in Arabidopsis by repressing DNA methylation genes. In addition to the previously described DREAM components these complexes also included the RBBP4 homolog MSI1, two LIN37 homologs as well as two uncharacterized proteins termed DREAM COMPONENT 1 (DRC1) and DREAM COMPONENT 2 (DRC2) (Ning et al, 2020). In addition, different plant homologs of LIN54, namely, TSO1, TCX2 (SOL2), TCX3 (SOL1), and TCX8, have been shown to be involved in different developmental processes, for example, in reproductive development (Liu et al, 1997; Hauser et al, 2000; Song et al, 2000; Andersen et al, 2007; Sijacic et al, 2011), the formation of stomata (Simmons et al, 2019) and tracheary elements (Clark et al, 2019), as well as senescence (Noh et al, 2021). However, the formation of a DREAM complex remained hypothetical in these contexts. Moreover, our knowledge on plant DREAM composition is still patchy and for instance, pairwise interaction assays among the identified components have only been performed to a limited extent (Ning et al, 2020).

Unraveling the RBR1 interactome upon DNA damage, we have identified here homologs of all core DREAM complex components known from humans and animals. By analyzing the composition of the DREAM complex under DNA damage as well as several other growth-modifying conditions, we have obtained a robust atlas of DREAM complex composition in Arabidopsis and have systematically mapped the interaction network of its constituents by binary interaction assays. Furthermore, we show that the SOG1 homolog NAC044 interacts with RBR1 in an LxCxE motif-dependent manner and functions in conjunction with the DREAM complex to suppress growth upon DNA damage.

# Results

## Identification of a plant complement of the animal DREAM complex proteins

To explore the RBR1 interactome upon DNA damage, we performed tandem affinity purification (TAP) assays (Van Leene et al, 2015) using cell cultures expressing N-terminally tagged RBR1 as bait. For DNA damage induction, we added the genotoxin cisplatin, a DNA cross-linker, 16 h before harvest. The experiment was performed in duplicate and resulted in the identification of 16 interactors, 15 of which passed the background threshold in both assays (Fig 1 and Tables S1 and S2). Notably, homologs of most components of the animal DREAM complex were found, including a protein with homology to LIN52, a DREAM component previously not recognized as such in plants (Kobayashi et al, 2015; Ning et al, 2020), and that we therefore named LIN52A (AT2G45250; Fig S1). To verify these interactions, reciprocal TAPs were performed in duplicates, again after 16 h of cisplatin treatment, taking N- and C-terminal fusions of TCX5 and LIN52A as bait proteins. This approach led to the identification of additional DREAM components, such as LIN37 and MYB3R proteins, as well as a second homolog of LIN52 (AT4G38280), named LIN52B, thus resulting in an entire equivalent of the mammalian DREAM complex (Fig 1 and Tables 1, S1, and S2).

Except for RBR1, Arabidopsis contains more than one homolog of each DREAM component, and for some components different family members were identified in our TAP experiments suggesting the existence of several versions of the DREAM complex after DNA damage treatment. However, our data also indicate specificity because only two of the eight Arabidopsis LIN54 homologs were identified, that is, TCX5 and TCX6. Whereas TCX5 was detected in all five experimental setups, TCX6 was found only in the one using N-terminally tagged LIN52A as a bait. For RBBP4, there are five homologs in Arabidopsis, called MSI1 to MSI5, but only MSI1 was identified in our experiments, yet consistently in all five assays. For LIN52, two homologs were co-purified, also here with a clear bias for one member, LIN52A, which was detected in five of five assays, whereas LIN52B was only found when N-terminally tagged TCX5 was used as bait. Whereas the Arabidopsis LIN9 homologs ALY1 and ALY2 were only present in the RBR1 and TCX5 TAP experiments, ALY3 was additionally found in both LIN52A experiments. We also found MYB3R3, which was previously shown to act as a repressive transcription factor, as well as MYB3R1, which has both activating and repressive functions, in the TCX5 TAP experiments. In addition, we

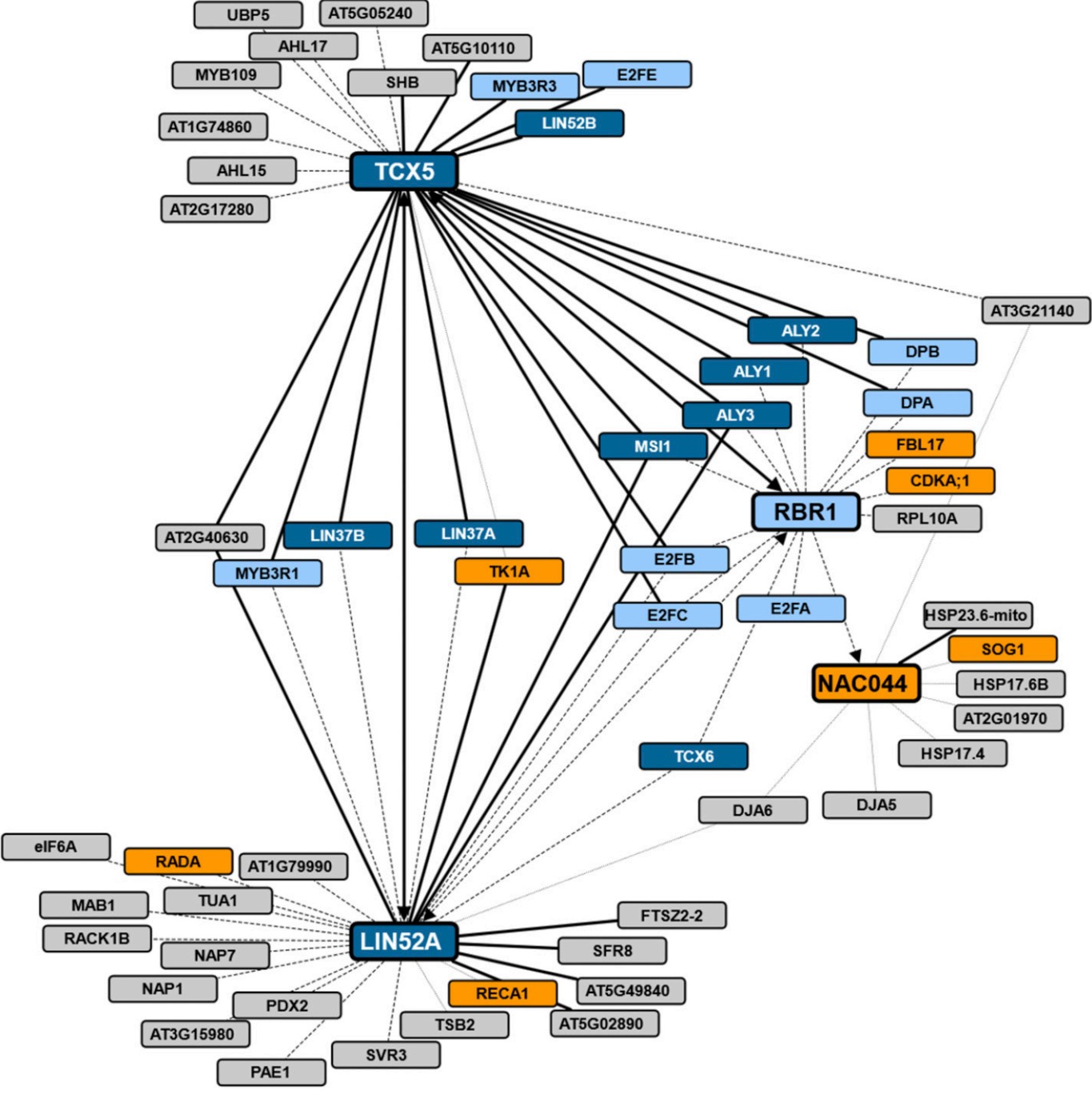

**Figure 1. Overview of tandem affinity purification (TAP) results from cisplatin-treated cell cultures.**

Cytoscape representation of all TAP experiments from cisplatin-treated cell culture. Proteins taken as baits are shown in large rectangles, proteins only found as prey are represented by small rectangles. TAPs with NAC044, LIN52A, and TCX5 as bait were performed with both N-terminally and C-terminally tagged proteins. Thick black edges indicate that the corresponding prey was detected with both tags, whereas dashed and dotted edges denote detection only with N-terminal and C-terminal tagging, respectively. For RBR1, only an N-terminally tagged version was used. If an edge connects two proteins which both served as bait in different experiments, arrowheads indicate which proteins have been found as prey. Dark blue: MuvB-core proteins; light blue: other DREAM components; orange: known DNA damage regulators; grey: other interactors.

observed apparent specificity in the family of E2F transcription factors. Whereas E2FB and E2FC were found in complex with RBR1, TCX5, and LIN52A as bait, E2FA was only found with RBR1 and the atypical E2FE (DEL1) only with TCX5.

In previously performed GFP-pulldown experiments using MYB3R proteins as baits, the key cell cycle kinase CDKA;1 was co-purified and thus suggested to be part of plant DREAM-like complexes (Kobayashi et al, 2015). Consistently, we also found CDKA;1 when we used RBR1 as

**Table 1. Arabidopsis sequence homologs of DREAM components and their presence in different affinity purifications.**

| Human DREAM Type | AGI | Alias | TAP results from cisplatin-treated cell culture | | | | | FP-IP results from seedlings | | | | | |
|---|---|---|---|---|---|---|---|---|---|---|---|---|---|
| | | | TAG-RBR1 | TAG-LIN54A | LIN54A-TAG | TAG-LIN52A | LIN52A-TAG | E2FA-GFP | E2FB-GFP | E2FC-GFP | RBR1-GFP | DPA-GFP | DPB-3xCFP |
| MYBL2 | AT4G32730 | MYB3R1 | | x | x | x | | | | | | | |
| | AT5G00540 | MYB3R2 | | | | | | | | | | | |
| | AT3G09370 | MYB3R3 | | x | x | | | | | | | | |
| | AT5G11510 | MYB3R4 | | | | | | | | | | | |
| | AT5G02320 | MYB3R5 | | | | | | | | | | | |
| E2F | AT2G36010 | E2FA | X | | | | | x | | | x | x | x |
| | AT5G22220 | E2FB | X | x | x | x | | | x | | x | x | x |
| | AT1G47870 | E2FC | X | x | x | x | | | | x | x | x | x |
| | AT5G14960 | E2FD/DEL2 | | | | | | | | | | | |
| | AT3G48160 | E2FE/DEL1 | | x | x | | | | | | | | |
| | AT3G01330 | E2FF/DEL3 | | | | | | | | | | | |
| DP | AT5G02470 | DPA | X | x | | | | x | x | x | x | x | |
| | AT5G03415 | DPB | X | x | x | | | x | x | x | x | | x |
| RBL | AT3G12280 | RBR1 | X | x | x | x | | x | x | x | x | x | x |
| *LIN9* | *AT5G27610* | *ALY1* | X | x | x | | | | x | x | x | | x |
| | *AT3G05380* | *ALY2* | X | x | x | | | | x | x | | | x |
| | *AT3G21430* | *ALY3* | X | x | x | x | x | x | x | x | x | x | x |
| *LIN54* | *AT3G22780* | *TSO1* | | | | | | | | | | | |
| | *AT4G14770* | *TCX2/SOL2* | | | | | | | | | | | |
| | *AT3G22760* | *TCX3/SOL1* | | | | | | | | | | | |
| | *AT3G04850* | *TCX4* | | | | | | | | | | | |
| | *AT4G29000* | *TCX5/LIN54A* | X | x | x | x | x | x | x | x | x | x | x |
| | *AT2G20110* | *TCX6/LIN54B* | | | | x | | | | | | | |
| | *AT5G25790* | *TCX7* | | | | | | | | | | | |
| | *AT3G16160* | *TCX8* | | | | | | | | | | | |
| *LIN37* | *AT1G04930* | *LIN37A* | | x | x | x | | | x | x | x | | x |
| | *AT2G32840* | *LIN37B* | | x | x | x | | | | x | | | x |
| *LIN52* | *AT2G45250* | *LIN52A/DRC1* | X | x | x | x | x | | x | x | x | x | x |
| | *AT4G38280* | *LIN52B* | | x | | | | | | | | | |
| *RBBP4* | *AT5G58230* | *MSI1* | X | x | x | x | x | | x | x | x | x | x |
| | *AT2G16780* | *MSI2* | | | | | | | | | | | |
| | *AT4G35050* | *MSI3* | | | | | | | | | | | |
| | *AT2G19520* | *MSI4* | | | | | | | | | | x | |
| | *AT4G29730* | *MSI5* | | | | | | | | | | | |

This table summarizes which Arabidopsis sequence homologs of known DREAM components have been identified by different complex purification approaches from different biological materials. For quantitative information, see Tables S2–S4. MuvB-core candidates are written in italics. Homologs which have not been significantly enriched in any of our experiments are written in grey.

a bait. However, CDKA;1 was not identified when we performed the experiment with one of the MuvB-core components tagged, that is, TCX5 or LIN52A. It was previously found that CDKB1 kinases have partially overlapping functions with CDKA;1, with CDKB1 playing a specific role during DNA damage (Nowack et al, 2012; Weimer et al, 2016). However, besides CDKA;1, we never retrieved any of the other 12 CDK proteins in our TAP experiments.

To corroborate our findings on the composition of the DREAM complex in plants, we performed pulldown experiments using young seedlings of Arabidopsis plants, where either RBR1 or one of the three E2F proteins tagged with GFP at their C-termini were expressed under the control of their own promoters (Kobayashi et al, 2015; Horvath et al, 2017). We identified and quantified proteins associating with E2FA-GFP, E2FB-GFP, E2FC-GFP, and RBR1-GFP against proteins interacting with the control GFP alone in six replicates each using label-free mass spectrometry (Lokdarshi et al, 2020). To certify interactors with statistical confidence, we computed the false discovery rate (FDR) and the amount ratio between proteins identified with the relevant baits versus GFP-only pull-downs and established thresholds as visualised in volcano plots (Fig S2A–D). The results show that among the proteins that co-precipitated with RBR1, E2FB, and E2FC from seedling extracts, we indeed found MuvB-core proteins, whereas these were completely absent from the E2FA pulldown experiments. Besides the DREAM components, we could also identify other interactors specific to RBR1 and the different E2F family members, suggesting functions linked to RNA binding and translational control (Lokdarshi et al, 2020) as well as chromatin organization (Table S3). Although our results are in accordance with previous experiments showing that homologs of the MuvB-core components LIN54 and LIN9 can be co-precipitated with E2FB-GFP and E2FC-GFP but not E2FA-GFP (Kobayashi et al, 2015), we additionally find homologs of LIN52 and RBBP4 in complex with E2FB and E2FC, indicating the presence of a complete set of DREAM proteins not only in our cisplatin-treated cell culture but also in untreated seedlings.

With the aim to comprehensively identify recurring interactors of RBR1 and the E2F/DP modules in seedlings, we performed a meta-analysis on a large set of pulldown mass spectrometry data generated from C-terminally GFP or CFP-tagged RBR1, E2FA, E2FB, E2FC, DPA, and DPB lines under a variety of growth-promoting and growth-restricting environmental conditions. In total, 182 IPs were performed and analyzed (50 GFP, 35 E2FA, 39 E2FB, 20 E2FC, 32 RBR1, 3 DPA, and 3 DPB). EdgeR statistical analyses were used on the spectral counts of all pulldown data and 217 preys passed a threshold of eight fold change (FC) and a p of 0.05 (Table S4). Fig 2 displays a reduced dataset as only prey proteins which were enriched in at least one-third of the IPs of one bait are shown (105 proteins). The vast size of the interactome likely reflects the multiple functions of RBR1 and E2F/DP. For example WIN2, PDF2.2, RIN4, and MOS1 (Table S4) might provide a molecular link to pathogen response, which is interesting in the light of the finding that the RBR1/E2F pathway has been shown to control programmed cell death in plant immunity (Wang et al, 2014). In the following, we focus on interactions relating to the DREAM complex.

The set of potential DREAM complex subunits found in seedlings under varying conditions largely matched what we observed in the cisplatin-treated cell culture (Tables S1 and S4). While LIN52A, TCX5,

LIN37A/B, ALY1/2/3, MSI1, DPA/B, E2FA/B/C, and RBR1 were identified in both approaches, only LIN52B and E2FE (DEL1) were found specifically in the cisplatin-treated cell culture. However, these proteins were not present in the RBR1 TAP experiment but only when the MuvB-core components were used as bait. With respect to the Arabidopsis LIN9 homologs, there is a clear bias in the frequency of experiments by which ALY3 was found with respect to the other two ALYs, with ALY3 clearly being the most prominent. Apart from what is shown in Fig 2 and Table S4, CDKA;1, MSI4, TCX6, as well as MYB3R3 and MYB3R1 were also present in some pulldown experiments from seedlings, but did not pass the more stringent selection criteria applied for the meta-analysis. With respect to E2FA, our data consistently showed that this E2F homolog is never found in complex with MuvB-core components in vivo, neither in the cisplatin-treated cell culture nor in seedlings grown under different conditions, whereas it could be readily co-purified with DP and RBR1.

Interestingly, a couple of proteins which appear to be unrelated to the animal DREAM complex were found in the TAPs of the MuvB-core components as well as in our FP-pulldown approaches, that is, the uncharacterized protein AT2G40630 (DRC2), which was found previously in complex with MSI1, LIN37B, and ALY3 (Derkacheva et al, 2013; Ning et al, 2020) and seems plant-specific, the EUKARYOTIC INITIATION FACTOR 6A (eIF6A), as well as a protein involved in pyridoxine biosynthesis (PDX2) (Fig 1 and Tables S1 and S4). Although beyond the scope of this investigation, it will be interesting to see whether and if so how they relate to DREAM function.

## A binary interaction atlas of the plant DREAM complex and implications of the LxCxE motif

To complement our data derived from the different complex isolation approaches, we performed yeast two-hybrid (Y2H) assays to establish an atlas of binary interactions for all of the here-identified Arabidopsis DREAM components (Fig 3A and B). In general, we see differences for a given protein pair depending on which interaction partner is fused to the activation (AD) or DNA-binding domain (BD), most likely due to folding differences of the different fusion proteins and 3D assembly of the reconstituted transcription factor. Interestingly, we also see differences in the interaction matrix between the different homologs, which might indicate the preferential formation of certain complex variants, although we cannot exclude a Y2H bias. In the following, we interpret our matrix as showing binding potential and describe the maximally observed interactions for each DREAM component. For more detailed, homolog-specific information, please refer to Fig 3A and B.

Considering the five members of the MuvB-core complex (ALY, LIN37, LIN52, TCX, and MSI), we found that LIN37 interacts with every component except ALY, whereas TCX binds RBR1, LIN37, and ALY in our Y2H system. Further, ALY associates with LIN52 and MSI, adding to an intricate interaction network among the plant MuvB-core members. Consistent with previous experiments, we found that the typical E2F transcription factors directly bind DP and RBR1 (Kosugi & Ohashi, 2002; del Pozo et al, 2006; Boruc et al, 2010; Magyar et al, 2012, 2005). In addition, our Y2H data indicate that contact of typical E2Fs to the MuvB-core occurs likely via LIN37, whereas the only MuvB-core RBR1 interaction interface seems to be on TCX. Notably,

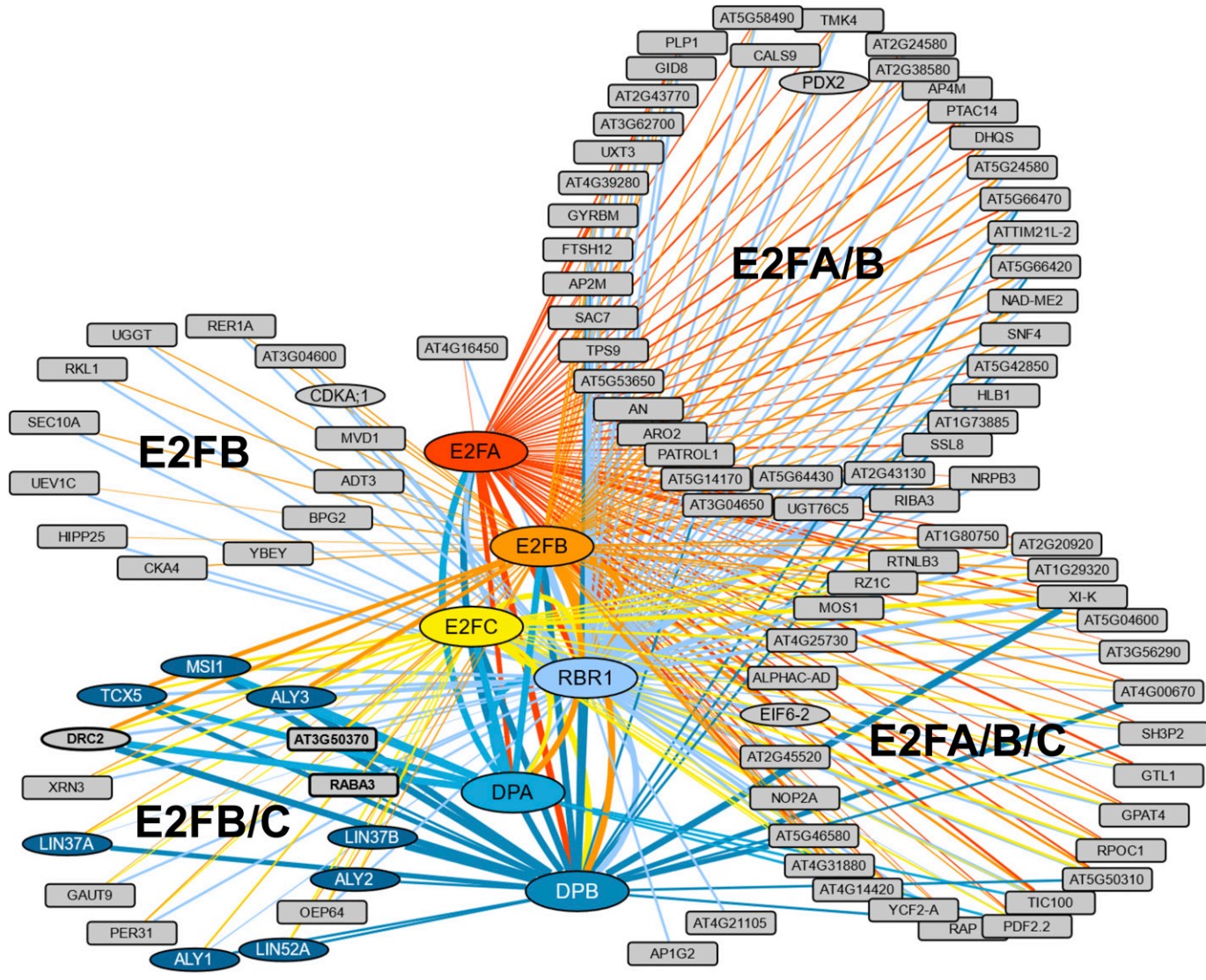

**Figure 2.  Meta-Analysis of FP-IP results from seedlings.**
Cytoscape representation of a meta-analysis of 182 FP-IPs using different bait proteins (35 × E2FA-GFP [blue], 39 × E2FB-GFP [red], 20 × E2FC-GFP [green], 32 × RBR1-GFP [yellow], 3 × DPA-GFP [orange], and 3 × DPB-CFP [pale-orange], 50 × GFP [control]). To reduce complexity, only prey proteins which were enriched in at least one third of the IPs of one bait are shown. For the more comprehensive dataset, see Table S4. The thickness of the edges corresponds to the frequency of positive IPs by which a bait/prey interaction was found. Proteins that were also identified in the tandem affinity purification experiments are represented by an ellipse. Prey proteins are grouped according to their occurences in different E2F-IPs. DREAM complex homologs are shown in dark blue. Additional proteins that display an interaction pattern like the DREAM component homologs, that is, do not interact with E2FA, but interact with E2FB/C and the DPs, are shown in bold with a thick outline.

the E2F dimerization partner DP is capable of binding both TCX and LIN37. As expected, the atypical E2F transcription factor E2FE (DEL1) does neither bind DP nor RBR1 but it is capable of interacting with three MuvB-core components, that is, LIN37, TCX, and LIN52. In contrast to the rather distinct association of the E2F/DP-RBR1 module with the MuvB-core, interaction of the latter with the MYB3R proteins seems to occur via multiple interfaces because we see interaction of MYB3R with all five core proteins, albeit of different strength, in the Y2H analyses. Finally, dimerization seems to be frequent among plant DREAM members, as found it not only for MYB3R3, but also for LIN37, LIN52, and TCX. In addition dimerization is also seen for the atypical E2FE.

While our pulldown data clearly show the existence of complete DREAM complexes in Arabidopsis when compared to the human version, protein–protein contact points within these complexes likely have shifted, as exemplified by the RBR1 MuvB-core interface. Pocket proteins like pRb contain a region called LxCxE binding cleft, which is bound by proteins displaying a signature similar or identical to the so-called LxCxE motif. In the human DREAM complex, the LxCxE binding cleft of p107 is bound by LIN52 (Guiley et al, 2015). However, the LxSxExL motif in HsLIN52, which is responsible for this interaction, is not conserved in the Arabidopsis LIN52 homologs (Fig S1), and when tested by Y2H assay, no direct interaction between RBR1 and LIN52A or LIN52B could be found

## A

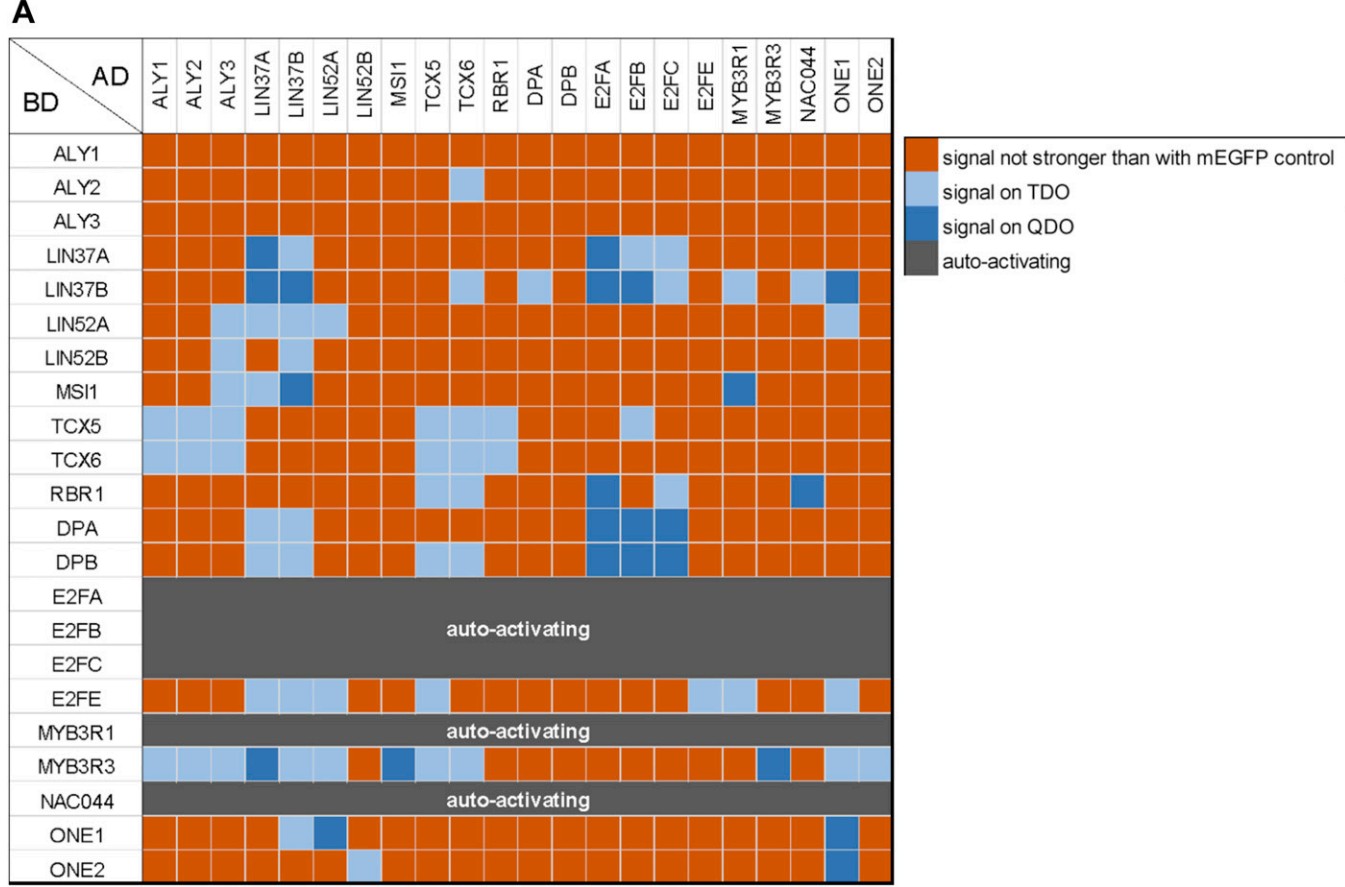

## B

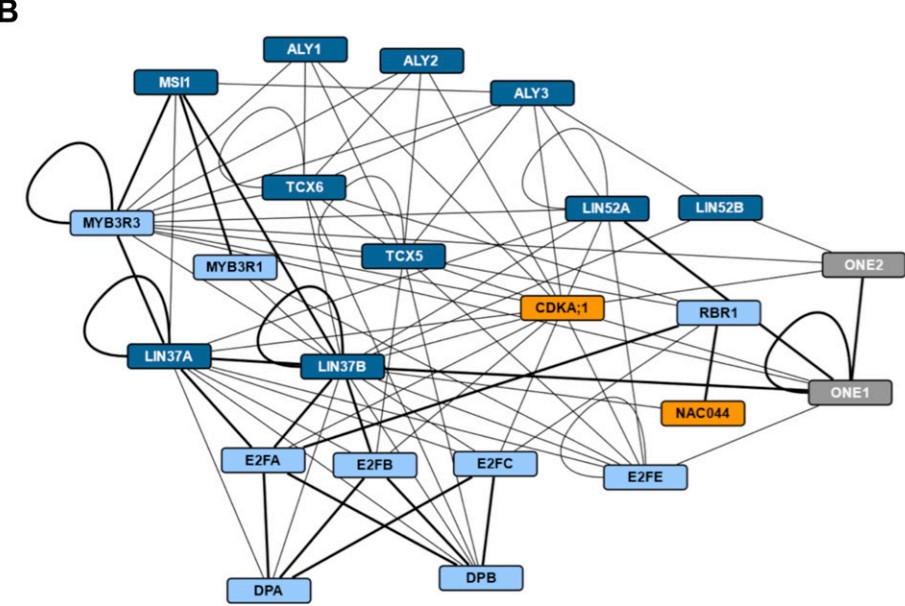

**Figure 3.   Binary interactions of DREAM complex components and additional proteins.**
Results of Y2H assays testing the here-identified DREAM complex components and selected additional proteins for binary interactions. AD, activating domain; BD, DNA-binding domain. **(A)** Interaction matrix. Signal strength was classified according to yeast growth on different dropout media in two categories and is indicated by shades of blue. Dark blue, signal on QDO; light blue, signal on TDO but not on QDO; orange, signal not stronger than with mEGFP control; dark grey, strong auto-activation observed using the BD construct and an AD-mEGFP control. **(B)** Cytoscape representation of the observed interaction network. Interactions are indicated by an edge between two protein nodes and were classified according to yeast growth in two categories. If yeast growth was observed with a pair of proteins in both AD/BD combinations, the stronger signal is shown. Thick line, growth on QDO; thin line, growth on TDO but not on QDO. Dark blue, MuvB-core proteins; light blue, other DREAM components; orange, known DNA damage regulators; grey, other interactors.

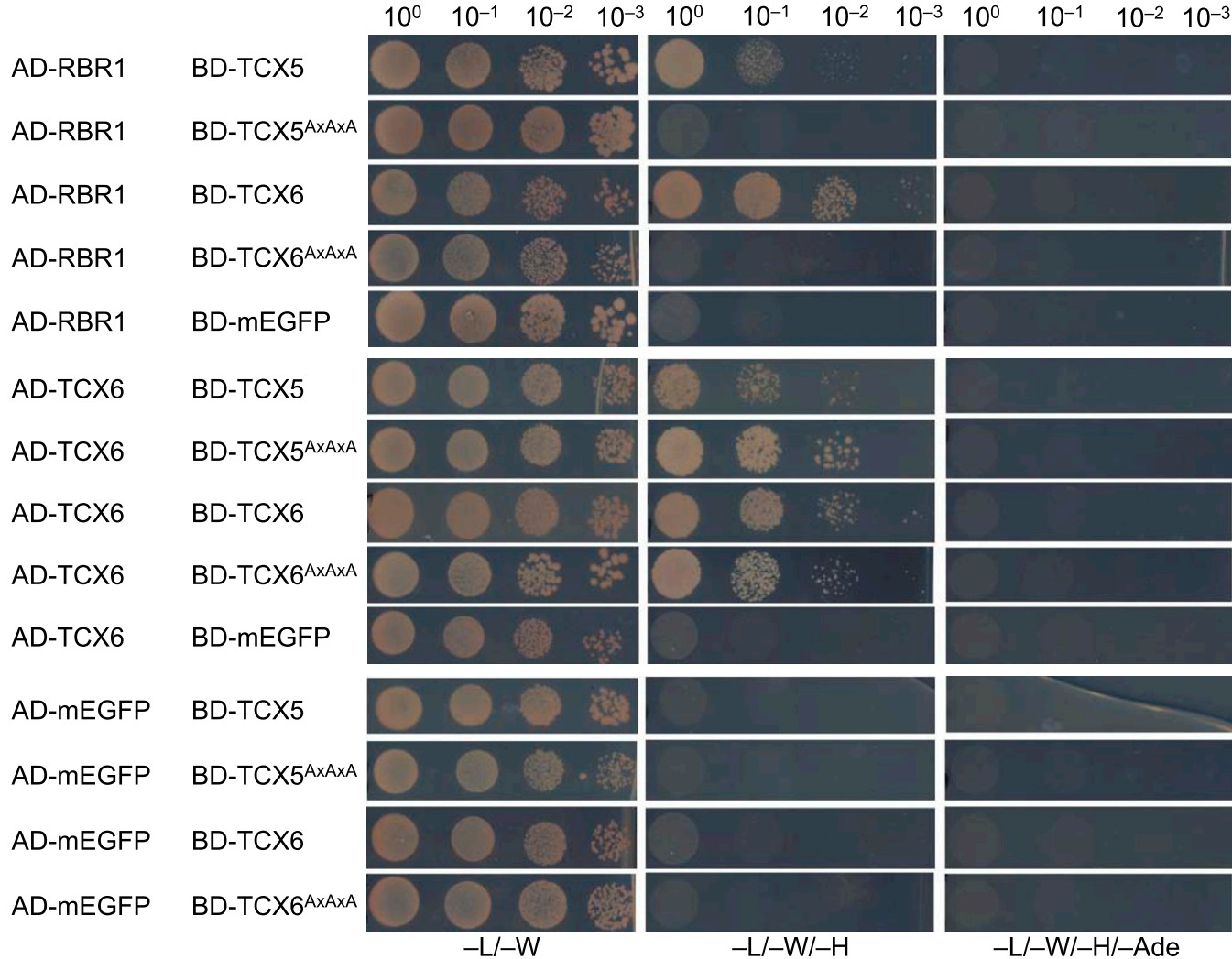

**Figure 4.  The LxCxE motif of TCX5 and TCX6 is essential for interaction with RBR1.**
Y2H interaction assays to test for binary interaction of wild type as well as mutant TCX5 and TCX6 with RBR1, TCX6, and mEGFP as auto-activation control. AxAxA replaces the LxCxE motif in the mutant proteins. AD, activating domain; BD, DNA-binding domain. Yeast cells were diluted as shown on top and spotted on different dropout media as indicated below. Growth on TDO and QDO indicates interaction.

(Fig 3A). Supporting this notion, Arabidopsis TCX5 and TCX6, the two LIN54 homologs identified in our pulldown experiments and inter-actors of RBR1 in the Y2H assays, both contain an LxCxE motif, which is not present in their human counterpart (Fig S3). When we mutated this motif in TCX to AxAxA, the interaction with RBR1 was abolished, whiereas dimerization with TCX6 was still possible (Fig 4), indicating that loss of RBR1 binding was not due to complete misfolding of the TCX proteins but dependent on a functional LxCxE motif.

Interestingly, not all LIN54 homologs in Arabidopsis carry an LxCxE motif and the region surrounding the motif in TCX5 is only conserved in TCX6 and TCX7 (Fig S3) (Andersen et al, 2007). Con-sistently, when testing non-LxCxE-bearing homologs, for example, TCX2 (SOL2), TCX3 (SOL1), and TSO1 in combination with RBR1 in the Y2H system, we could not detect any interaction while binding assays with the MYB3R3 transcription factor were positive, indi-cating that lack of yeast growth in combination with RBR1 was not due to a technical problem with the TCX2, TCX3, and TSO1 constructs

per se (Fig S4). Furthermore, none of the LIN54 homologs without LxCxE were found in any of our pulldown experiments although, according to publicly available transcriptome data, at least some of them are well expressed in seedlings and cell culture (Andersen et al, 2007) (Fig S5A and B).

## NAC044 links the RBR1 interactome to DNA damage

Apart from DREAM complex components, we also found the tran-scriptional regulator NAC044 in our RBR1 TAPs performed with cisplatin-treated cell culture (Fig 1 and Tables S1 and S2). NAC044 is a close homolog and transcriptional target of the major DNA damage regulator SOG1 and has recently been shown to limit root growth after DNA damage (Takahashi et al, 2019). However, how the latter is achieved molecularly is still unknown.

When we performed reciprocal TAP experiments using N- and C-terminally tagged versions of NAC044 as bait, we identified SOG1

**Figure 5. The LxCxE domain of NAC044 is essential for interaction with RBR1.**
Y2H interaction assays to test for binary interaction of wild type as well as mutant NAC044 with RBR1, LIN37B, and mEGFP as auto-activation control. AxAxA replaces the LxCxE motif in the mutant NAC044. AD, activating domain; BD, DNA-binding domain. Yeast cells were diluted as shown on top and spotted on different dropout media as indicated below. Growth on TDO and QDO indicates interaction.

as an interactor of C-terminally tagged NAC044, whereas none of the DREAM components were co-precipitated at a significant level. However, RBR1 could be identified in the background when using C-terminally tagged NAC044 as bait corroborating their interaction (Table S2). In addition, common interactors of NAC044 and LIN52A as well as NAC044 and TCX5 were found in the affinity purifications, that is, AT2G22360 (DJA6) and AT3G21140, respectively (Tables S1 and S2 and Fig 1). Since AT3G21140 has been so far uncharacterized, we named it ONEIRIC 1 (ONE1; oneiric meaning "relating to dreams or dreaming"), reflecting its potential relation to the DREAM complex.

To more directly assay if NAC044 might be part of an extended DREAM complex, we performed Y2H interaction assays between NAC044 and all DREAM components identified in this project as well as the common interactor ONE1 and its close homolog ONE2 (AT1G51560) (Fig 3A and B). Whereas NAC044 fused to the Gal4 DNA-binding domain (BD-NAC044) was auto-activating and could not be analyzed in the Y2H assays, NAC044 fused to the activation domain (AD-NAC044) strongly interacted with RBR1 and at moderate strength with LIN37B. The interaction with SOG1 could not be tested in Y2H, because SOG1 was also auto-activating when fused to the DNA-binding domain. Intriguingly, NAC044 also contains an LxCxE motif located at amino acids 303–307 and we wondered if this was relevant for RBR1 interaction. Therefore, we generated a NAC044 version in which the LxCxE motif was changed to AxAxA and re-tested the respective mutant for interaction with RBR1 as well as LIN37B (Fig 5). Whereas RBR1 interaction was clearly abolished in the mutant, the interaction with LIN37 remained, indicating that the induced mutation did not lead to a completely misfolded protein but specifically abolished RBR1 binding.

In the Y2H system, ONE1 showed interactions with LIN37B, LIN52A, E2FE, MYB3R3, and the ability to homodimerize as well as heterodimerize

with its homolog ONE2. ONE2 on the other hand is bound weakly by LIN52A and MYB3R3 as well. Although these results are in favor of an in vivo involvement of ONE1 and ONE2 with DREAM complex components, a direct interaction with NAC044 could not be demonstrated by Y2H.

**Dynamics of NAC044 and selected DREAM components after DNA damage**

*NAC044* has been shown to be a transcriptional target of SOG1 and its mRNA accumulates upon DNA damage (Bourbousse et al, 2018; Takahashi et al, 2019). To monitor protein amount and localization with and without DNA damage, we generated a genomic reporter where the coding sequence of *mEGFP* was inserted right before the start codon of *NAC044*. This construct was considered functional as it complemented the *nac044-1* growth phenotype upon cisplatin treatment (Fig S6A–D). When we analyzed root tips of the reporter line grown on control plates for protein expression, we did not find any mEGFP signal in most of the cells. However, occasionally we saw strong nuclear accumulation of mEGFP-NAC044 in isolated cells in different tissue layers of the root. Next, we monitored NAC044 abundance upon treatment with 50 µM cisplatin. Time point zero corresponds to what we observed in untreated roots, that is, very few, apparently randomly located cells with clear nuclear fluorescence signal (Fig 6A). Beginning at 6–8 h after treatment onset, we saw enhanced nuclear mEGFP accumulation that reached a maximum after 24 h and could be observed in nearly all cells of the root tip. However, the fluorescence intensity was very different between different cells, resulting in a salt-and-pepper-like pattern (Fig 6A). Consistent with the observation that NAC044 protein is largely absent from roots in non-stressed conditions, we never found NAC044 in any of the pulldown experiments from seedlings (Table S4).

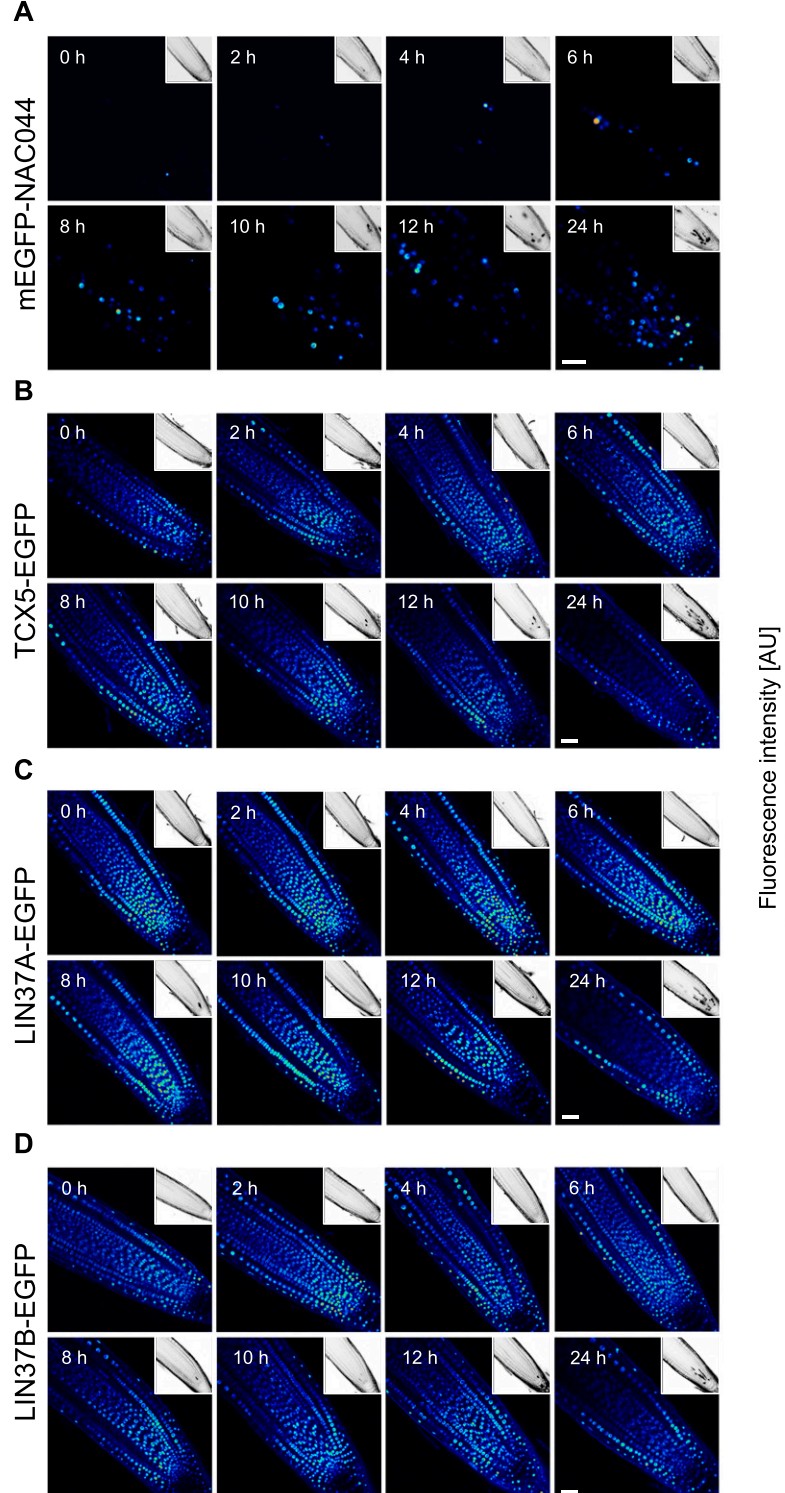

**Figure 6. Time course of NAC044, TCX5, LIN37A, and LIN37B protein expression after cisplatin treatment.** **(A, B, C, D)** Using genomic reporter lines to include the native regulatory sequences, expression of mEGFP-NAC044 (A), TCX5-EGFP (B), LIN37A-EGFP (C), and LIN37B-EGFP (D) was followed for 24 h after transfer of 6-d-old seedlings to 50 μM cisplatin-containing plates. Representative images of root tips at different time points, as indicated in the upper left corner of each panel, are shown here using the royal LUT in ImageJ. PI staining of the corresponding part of the root tip is shown as inset in the upper right corner of each panel. Scale bar = 30 μm. Microscopic settings were kept constant for each line, but not necessarily between lines.

Regarding the dynamic pattern of NAC044 after DNA damage, we wondered if also other components of the RBR1 interactome, specifically the DREAM core components, would change expression after DNA damage. We therefore generated genomic TCX5-EGFP, LIN37A-EGFP, and LIN37B-EGFP reporter lines and analyzed the expression of these reporters after treatment with 50 μM cisplatin. For the first 12 h, TCX5 as well as LIN37A and LIN37B showed clear nuclear signals in most cell files of the root tip, without obvious dynamics (Fig 6B–D). At 24 h, expression in the central cylinder appears slightly decreased, however, is still present. Thus, we

**A**

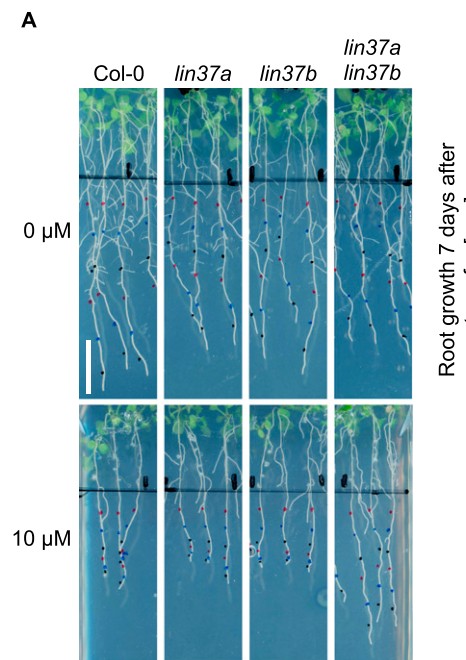

**B**

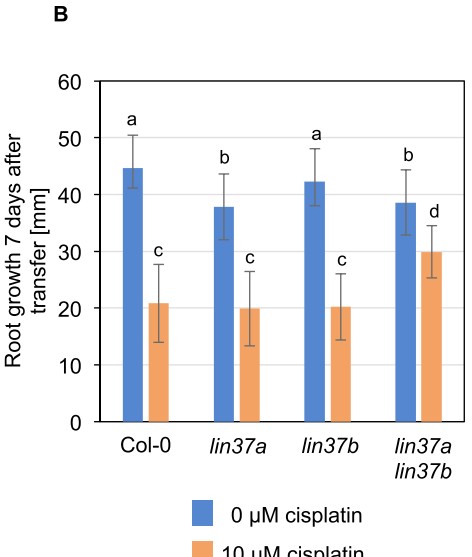

0 µM cisplatin
10 µM cisplatin

**Figure 7. Double mutants for *LIN37A* and *LIN37B* display less repression of root growth under DNA-damaging conditions than the wild type.**
**(A, B)** Root growth of the wild type, *lin37a*, *lin37b*, and *lin37a lin37b* double mutants on control plates and in the presence of cisplatin (A: representative pictures; B: quantification). Plants grown for 5 d in the absence of cisplatin were transferred to medium containing 0 or 10 µM cisplatin, and grown for further 7 d. Data are presented as mean ± SD (n = 10). Significant differences as determined by two-way ANOVA and Tukey–Kramer post hoc test ($P < 0.05$) are indicated by differing letters over the bars.

conclude that, although the expression dynamics of NAC044 and MuvB-core components differ upon DNA damage, they are expressed in overlapping patterns and therefore, likely engage with the same pool of RBR1 in a cell type- and/or cell cycle phase-specific manner.

### Loss of DREAM components compromises growth arrest under DNA-damaging conditions

Because *nac044* mutants have been shown to grow better under DNA-damaging conditions than the wild type (Takahashi et al, 2019) (Figs S6 and S8), we wondered if the same would hold true for mutants in other components of the RBR1 DNA stress interactome, in particular the DREAM complex. Therefore, we first isolated T-DNA insertion mutants for several components of the MuvB-core and tested for gene expression. No full-length transcript was found in homozygous mutants of *ALY1* (*aly1-1*, *aly1-2*, *aly1-3*), *ALY2* (*aly2-2*, *aly2-3*, *aly2-4*), *ALY3* (*aly3-1*, *aly3-3*, *aly3-4*), *LIN37A* (*lin37a-2*), *LIN37B* (*lin37b-3*), *TCX5* (*tcx5-1*, *tcx5-2*), *TCX6* (*tcx6-1*), and *LIN52B* (*lin52b-1*) (Fig S7A–D). On the other hand, all T-DNA insertion lines tested for *LIN52A* showed full-length transcripts. Thus, we generated a CRISPR allele, *lin52a-c1*, to be used in our analysis (Fig S7C).

When root growth was analyzed under DNA-damaging conditions, the results between biological replicates were considerably variable, with some genotypes showing repeatedly better growth than the wild type upon cisplatin or mitomycin C (MMC) treatment, however not consistent enough to always yield statistically significant results (see Fig S8A–D for examples). The source of this variation is currently unknown. Because the *lin37* single mutants frequently showed better growth than the wild type on DNA-damaging media, we generated a *lin37a-2 lin37b-3* double mutant, aiming at an enhanced phenotype. Indeed, when we analyzed its root growth (Fig 7A and B), the double mutant consistently grew

significantly ($P < 0.05$) better than the wild type and the respective single mutants on 10 µM cisplatin-containing plates, even though on control plates, the double as well as the *lin37a* single mutant occasionally even displayed slightly shorter roots than the wild type. Because we previously identified ALY1 as an RBR1 target gene which is up-regulated under DNA-damaging conditions (Bouyer et al, 2018), we also aimed at a comprehensive mutant analysis of the ALY family. However, in this case, none of the double mutants showed significant growth difference on DNA-damaging media when compared with the wild type (Fig S8D), whereas the triple mutant could not be analyzed because of lethality (Ning et al, 2020).

In summary, by mutant analysis, we show that in *Arabidopsis thaliana,* the MuvB-core component LIN37 is functionally relevant to restrict root growth after DNA damage.

### E2FB is required for DNA damage-induced cell cycle arrest

In addition to the MuvB-core, we decided to also follow up the typical E2F transcription factors. Whereas E2FA (Horvath et al, 2017) and E2FC (Gómez et al, 2019) have previously been shown to be part of the DDR network, the role of E2FB has not yet been explored in detail. When we analyzed seedling roots expressing *pgE2FA-3xvYFP* or *pgE2FB-3xvYFP* by confocal microscopy after 24 h growth on cisplatin-containing or control plates, respectively, we observed that the overall E2FB signal seemed slightly enhanced after 24 h growth on genotoxin, whereas the E2FA signal was very similar to the signal in roots grown on control plates (Fig 8A).

For a more quantitative analysis of all three typical E2F transcription factors, we made use of our well-established E2F pulldown system from seedlings, that is, 6-d-old seedlings were incubated with and without 50 µM cisplatin for 24 h and subsequently analyzed by GFP-IP and mass spectroscopy. Fig 8B–D show the ratio of peptides found in cisplatin-treated versus

**A**

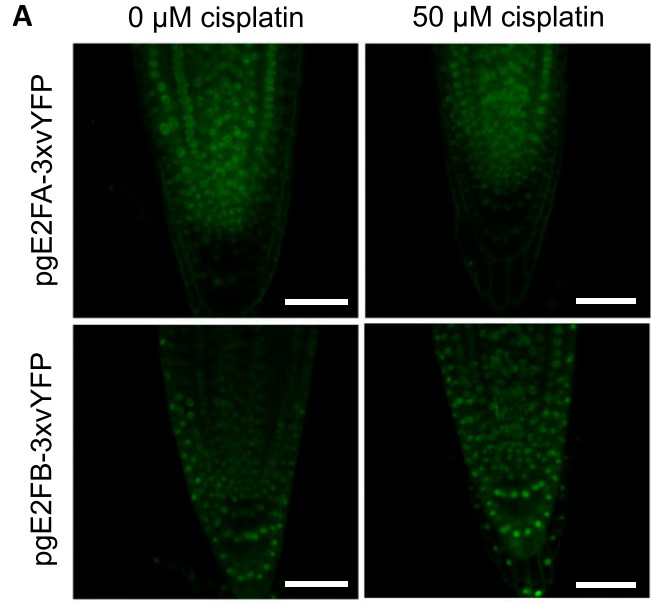

0 µM cisplatin    50 µM cisplatin

pgE2FA-3xvYFP

pgE2FB-3xvYFP

**B**

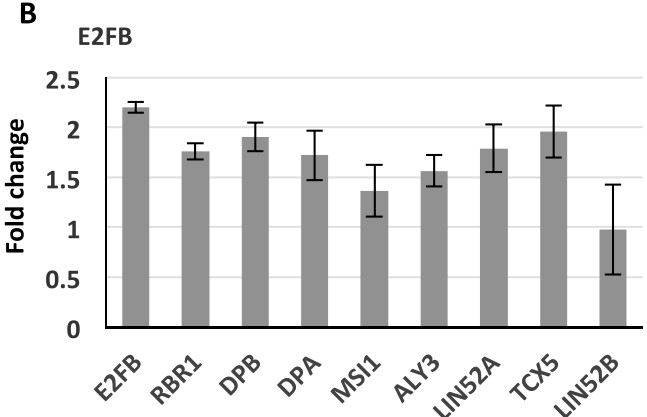

E2FB

Fold change

E2FB, RBR1, DPB, DPA, MSI1, ALY3, LIN52A, TCX5, LIN52B

**C**    **D**

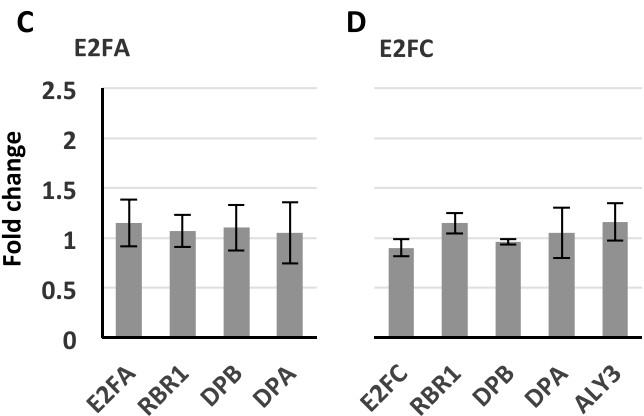

E2FA    E2FC

Fold change

E2FA, RBR1, DPB, DPA    E2FC, RBR1, DPB, DPA, ALY3

**Figure 8. An E2FB-containing DREAM complex is enriched under DNA damage conditions.**
**(A)** Confocal analysis of seedling roots expressing *pgE2FA-3xvYFP* or *pgE2FB-3xvYFP* indicates an overall increase in E2FB-3xvYFP but not E2FA-3xvYFP fluorescence after 24 h treatment with 50 µM cisplatin. Scale bar = 50 µm.
**(B, C, D)** Quantification by FP-IP. **(B, C, D)** 1 d of cisplatin treatment (50 µM) of 6-d-old E2F-GFP-expressing seedlings increases the quantity of DREAM

untreated plants. On a whole seedling level, the amount of E2FA was not significantly changed after DNA damage treatment, and also, we still did not find MuvB-core components in association with E2FA after incubation with cisplatin. Likewise, the amount of E2FC stayed constant after DNA stress treatment as did the load of co-precipitated RBR1, DP, and ALY3. In contrast, for E2FB and most of its interacting DREAM components, we saw an up to two-fold enrichment after 24 h of cisplatin treatment. Although these results indicate a global enrichment in E2FB-containing complexes after cisplatin treatment, we cannot exclude that different tissues of the seedling behave differently and some cell type-specific enrichment or depletion of the different complexes might not be revealed at this level of analysis. In addition, we would like to point out that most cells in a seedling are post-mitotic so that changes taking place in actively dividing cells are likely only reflected by rather minor changes in total protein amount.

To further zoom in on E2FB functionality, we analyzed *e2fb* mutants of different allelic strengths, *e2fb-1* and *e2fb-2* (Leviczky et al, 2019), by measuring root growth, cell cycle parameters, and cell death upon cisplatin treatment. As a first step, we monitored root growth after transfer to 15 µM cisplatin-containing plates or control plates. The two *e2fb* alleles have been shown to differ in phenotypic strength, such as embryo size and seed maturation, which could be related to the site of T-DNA insertion, resulting in the inclusion (*e2fb-1*) or exclusion (*e2fb-2*) of the dimerization domain within the truncated E2FB protein (Leviczky et al, 2019). As in seed maturation, the two mutant alleles showed a differential response to DNA damage, with the more severely affected *e2fb-2* mutants showing a significantly longer root ($P < 0.05$) than the wild type 6 d after transfer onto 15 µM cisplatin plates (Figs 9A and B and S9A for time course). This suggests that like the RBR1 interactor NAC044 and the MuvB-core component LIN37, E2FB is required for maximum inhibition of root growth upon DNA damage.

A cellular response to DNA damage is cell cycle arrest to allow time for repair of damaged DNA regions (Nisa et al, 2019). Thus, we asked whether E2FB plays a role to control G1-to-S transition. To this end, we transferred seedlings for 3 h onto 50 µM cisplatin plates and then carried out a 30 min EdU labelling of root tips to quantify meristematic cells in S phase as a percentage of the total number of nuclei stained with DAPI. Whereas in wild-type plants and *e2fb-1* mutants S phase count was reduced by 30–40%, the stronger *e2fb-2* allele did not show any significant change (Fig 9C and D). The DAPI staining also allowed us to count cells that undergo mitosis, which in our conditions were seen at a frequency of on average three to six in an optical section of the untreated root tip. Although none of the differences in mitotic cell count was statistically significant applying an ANOVA test, we still saw a trend (Fig S9B and C). As expected from previous publications (Weingartner et al, 2003) we observed a reduction in mitotic cells upon cisplatin treatment in

components specifically in the protein complexes containing E2FB (B) but not E2FA (C) or E2FC (D). Interacting protein partners were immunopurified from the corresponding E2F-GFP translational lines by using anti-GFP-containing magnetic beads, and components were identified with mass spectrometry. Graphs show fold change calculated as a ratio of immunoprecipitated components after cisplatin treatment relative to untreated conditions. Values represent the mean of three biological replicates ± SE.

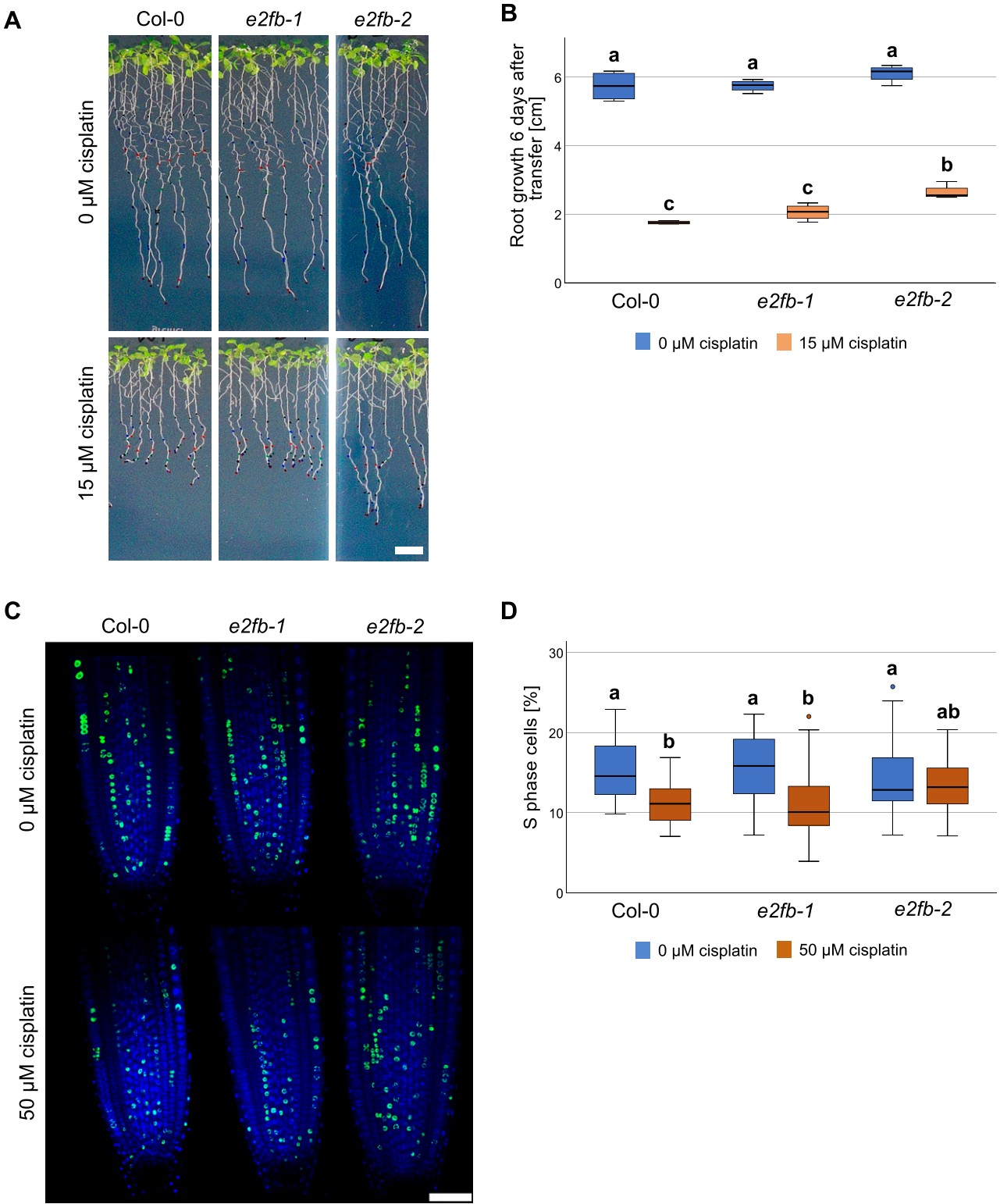

**Figure 9. DNA damage-induced cell cycle arrest requires E2FB.**
**(A)** Whole-plant photographs of Col-0, *e2fb-1*, and *e2fb-2* treated with 0.05% vol/vol DMF (mock) or 15 $\mu$M cisplatin. Scale bar = 10 mm. **(B)** Primary root length Col-0, *e2fb-1*, and *e2fb-2* seedlings after 6 d of 15 $\mu$M cisplatin or mock treatment. An average of 20 roots were measured for each genotype and condition. A box plot of the replicate means is shown. Significant differences were determined by ANOVA and Tukey post hoc test ($P < 0.05$). **(C)** Representative confocal images of EdU-labelled root tips of Col-0, *e2fb-1*, and *e2fb-2* treated with 50 $\mu$M cisplatin or 0.16% vol/vol DMF (mock) for 3 h. Scale bar = 50 $\mu$m. **(D)** Percentage of EdU-positive S phase cells relative to DAPI-stained nuclei in the root meristems of Col-0, *e2fb-1*, and *e2fb-2* treated with 50 $\mu$M cisplatin or 0.16% DMF (mock) for 3 h. An average of 25 roots were imaged for each genotype and condition. A box plot of the replicate means is shown, outlier values are shown as circles. Significant differences were determined by ANOVA and Tukey post hoc test ($P < 0.05$).

the wild type, indicating a G2/M cell cycle arrest because of checkpoint activation. In contrast, both the *e2fb-1* and *e2fb-2* mutants had an increased number of mitotic cells upon cisplatin treatment compared with the untreated control (Fig S9B). Thus, *e2fb* mutant cells either enter mitosis more readily than the wild type under DNA-damaging conditions and/or stay in this cell cycle phase for a prolonged time.

Irreparable DNA damage frequently results in cell death of rapidly dividing tissues to prevent damaged DNA being passed on to the daughter cells. To assess this cellular response, we carried out propidium iodide (PI) staining and quantified cell death area in the root tip 1 d after transfer to 50 µM cisplatin. Because cell death in the vasculature or in columella and lateral root cap initial cells (stem cells and their immediate daughters) is known to evolve differently in response to DNA damage, we quantitated these areas separately as described before (Horvath et al, 2017). Compared with the wild type, both *e2fb* mutants showed reduced cell death areas; however, the reduction was only statistically significant for columella and lateral root cap initial cells but not in the vasculature (Fig S10A–C).

Taken together, our data show that E2FB is needed to restrict root growth after DNA damage which is in part due to a function at G1/S transition and likely includes additional functions in the control of cell death and entry into mitosis.

# Discussion

The adjustment of gene expression is a key instrument in the DDR of cells and organisms. On the one hand, DNA repair genes have to be induced or cell death is triggered to eliminate excessively damaged cells. On the other hand, the expression of genes involved in cell proliferation and growth is reduced to provide time for repair.

Here, we have focussed on the Arabidopsis pocket protein RBR1 and its role in DNA damage control by elucidating RBR1's protein interaction network. A previous genome wide analysis of DNA sites occupied by RBR1 has revealed many cell cycle-related targets, reflecting its role in S- and M-phase control, but also sets of proteins involved in chromatin organization and DNA damage repair (Bouyer et al, 2018). However, in which protein assemblies RBR1 fulfills its multiple functions is only starting to be explored (for a recent review see Desvoyes and Gutierrez [2020]). To reveal with which proteins RBR1 cooperates upon DNA damage, we have identified here the RBR1 interactome of a cisplatin-treated cell culture. First of all, we found a whole-plant equivalent of the human DREAM complex, including two proteins which have sequence similarity to LIN52 and have therefore been named LIN52A (formerly called DRC1, Ning et al, 2020) and LIN52B. However, according to our Y2H data, the arrangement of the components in the Arabidopsis complex(es) seems different from the one in humans because the contact of the MuvB-core to RBR1 depends on an LxCxE motif in the plant LIN54 orthologs TCX5/TCX6 and not on a LIN52 LxSxExL motif as in humans. Remarkably, the RBR1-binding LxCxE motif in TCX5/TCX6 is not found in all members of the TCX family, but is part of a conserved region termed ALM motif (due to its amino acid sequence) that is characteristic for type 2 TCX proteins (Andersen et al, 2007) including TCX5, TCX6, and TCX7 from Arabidopsis. Interestingly,

the ALM motif with the consensus sequence SPxTxA**LMC**D**E** includes a conserved SP site preceding the LxCxE motif, i.e., a potential phosphorylation site for proline-directed kinases, such as CDKA;1, which was also present in the RBR1 TAP. For TCX6 and TCX7, this site has already been detected to be phosphorylated in vivo as documented in the Arabidopsis protein phosphorylation site database PhosPhAt4.0 (http://phosphat.uni-hohenheim.de) (Durek et al, 2010). It will thus be interesting to see in future if such a phosphorylation impacts DREAM complex assembly and if so, which kinase is involved. Furthermore, to our knowledge, only TCX proteins with the ALM motif have been found to interact with other MuvB-core proteins in planta (Kobayashi et al, 2015; Ning et al, 2020), although recently TCX8 has been found to bind to ALY3 in the Y2H system (Noh et al, 2021). So either the non-ALM motif-bearing LIN54 homologs of Arabidopsis are part of less stable or less abundant versions of the DREAM complex, making complex precipitation from plant material difficult or they do not build a DREAM complex at all but rather function as transcriptional regulators in different assemblies.

For mammalian cells, it has been shown that the MuvB-core forms different complexes depending on the cell cycle phase. In plants, the existence of activating and repressing DREAM complexes, depending on the MYB3R and E2F version involved, has been postulated (Fischer & Müller, 2017). Thus, it is well possible that an even larger variety, depending on the cellular context, of related complexes exists. For example, under DNA-damaging conditions, we have not only found DREAM components but also the transcription factor NAC044 to be part of the RBR1 interactome, and in Y2H assays, NAC044 directly binds to RBR1 and to the MuvB-core component LIN37. Because TCX5 and TCX6 as well as NAC044 interact with RBR1 in an LxCxE motif-dependent manner and thus, likely target the same site in RBR1, they are probably not part of the same complex unless this complex contains several copies of RBR1. It is also possible that NAC044 recruits the DREAM complex to certain promoters under DNA damage with its position in the complex being subsequently replaced by TCX5/TCX6. The exact composition and the dynamics of the one or multiple DREAM complexes need to be resolved in the future. However, our results already provide evidence that DREAM and NAC044-containing complexes are both involved in restricting growth after DNA damage and likely are functionally interdependent because both were found as part of the RBR1 DNA stress interactome in cell culture.

A dynamic composition of RBR1-containing complexes is also suggested by the different accumulation patterns of NAC044 versus DREAM components. In root tips, the expression pattern of LIN37B and TCX5 appears to be not altered upon DNA damage, with the exception of a reduction in protein amount in the vasculature after 24 h. In contrast, NAC044 shows a patchy pattern reminiscent of cell cycle-dependent regulation, and accumulates after DNA damage treatment over time, peaking at 24 h after exposure to cisplatin. Interestingly, it has been shown that NAC044 positively feeds back on the amount of MYB3R3, that is, whereas MYB3R3 mRNA levels remain stable, MYB3R3 protein accumulates in an NAC044/NAC085-dependent manner after DNA damage and results in a reduced expression of mitotic genes (Takahashi et al, 2019). This led to the hypothesis that the transcription factor NAC044 controls

protein abundance of MYB3R3 leading to the repression of mitotic genes, which, in addition to a direct interaction with RBR1 and LIN37, would represent a second route by which NAC044 impinges on DREAM function after DNA damage.

In animal cells, it has been shown that G1/S and G2/M genes are differentially regulated by different pocket proteins, that is, the non-DREAM complex-forming pRb and the DREAM-compatible p130 and p107, in response to DNA damage. According to a current model, G1/S genes are mainly repressed by pRb, with a contribution of p130 and p107, whereas p130 and p107 repress G2/M genes (Schade et al, 2019). In Arabidopsis, there is only one pocket protein, RBR1, by which we could co-precipitate an almost complete set of DREAM proteins after DNA damage. However, consistent with previous results (Horvath et al, 2017), we found that, in contrast to E2FB and E2FC, E2FA never seems to be incorporated in a DREAM complex in vivo, neither under DNA-damaging nor under control conditions. Considering that in our Y2H assays E2FA interacted not only with RBR1 but also with the DREAM core component LIN37, we suspect plant-specific posttranslational modifications or the binding of inhibitors to be responsible to prevent incorporation into a DREAM complex under in vivo conditions. Although not present in a DREAM complex, E2FA was still part of the RBR1 interactome upon DNA damage. Thus, we conclude that DREAM and non-DREAM pocket protein complexes are also present in plants after DNA damage.

Interestingly, we could also precipitate the atypical, repressive E2FE (DEL1) from DNA-stressed cells using TCX5 as a bait, suggesting the presence of yet another complex variety. In humans, the atypical, non-pocket protein-binding E2F7 is also involved in DDR as it mediates, for example, the transcriptional repression of indirect p53/TP53 target genes involved in DNA replication (Carvajal et al, 2012) and on the other hand negatively regulates genes involved in DNA damage repair (Mitxelena et al, 2018).

As it is well known in yeast and animal cells, upon DNA damage, cell cycle checkpoints are activated at the G1-to-S and G2-to-M transitions to allow time for DNA repair. This has also been shown to be the case in plants (Carballo et al, 2006). To zoom in on the DREAM function after DNA damage, we focussed here on E2FB because we found an E2FB-containing DREAM complex to be slightly but significantly enriched upon DNA damage and root growth was less reduced in e2fb mutants than in the wild type when treated with DNA-damaging agents. This indicates that E2FB is required for DNA damage-induced cell cycle checkpoints. We showed by visualising S-phase cells and mitotic cells within the root meristem after a short 3 h treatment with DNA-damaging drugs that these checkpoints operate at the G1-to-S and potentially also at the G2-to-M phase transition. This is consistent with E2FB's function to control both cell cycle transitions in cultured to-bacco cells (Magyar et al, 2005). Furthermore, RBR1 repression on E2FA was shown to regulate the DNA damage–induced cell death (Horvath et al, 2017). Our data now indicate that E2FB is also required for this process. Interestingly, a mutant phenotype similar to e2fb's with less cell death and impaired G2 arrest after DNA damage treatment was seen for nac044 nac085 double mutants (Takahashi et al, 2019). Albeit E2FB was originally described as an activator of cell cycle progression (Magyar et al, 2005) and also hypothesized to be part of activating DREAM complexes in plants (Kobayashi et al, 2015), we recently found that E2FB in association with RBR acts as a repressor of cell proliferation during leaf development (Őszi et al, 2020). Consistently, it has been shown that many transcription factors are able to fulfill activating

as well as repressive functions depending on the molecular context (Bauer et al, 2010; Boyle & Després, 2010).

In summary, our data indicate the existence of multiple routes of transcriptional control after DNA damage, that is, via E2FA-RBR1, E2FE-TCX5, and various DREAM complexes likely involving different homologs. The functional relevance of DREAM-like complexes in DNA damage is shown by the compromised root growth arrest of lin37 and e2fb mutants after damage, resembling the derepressed growth of nac044 mutants on genotoxic media. We propose that by direct interaction with RBR1 and LIN37B, NAC044 cooperates with the DREAM complex to suppress root growth after DNA damage by controlling cell cycle progression. Identifying NAC044 target genes and analyzing complex composition at tissue or cellular level will shed light on the molecular nature of this cooperation in the future.

# Materials and Methods

## TAP

Cloning of transgenes encoding N- or C-terminal GS[rhino] tag (Van Leene et al, 2015) fusions under control of the constitutive cauliflower tobacco mosaic virus 35S promoter and transformation of Arabidopsis cell suspension cultures (PSB-D) with direct selection in liquid medium was carried out as previously described (Van Leene et al, 2011). Cisplatin was added to a final concentration of 30 $\mu$M 16 h before harvest of the cell culture. TAP experiments were performed with 100 mg of total protein extract as input as described in Van Leene et al (2015). Bound proteins were digested on-bead after a final wash with 500 $\mu$l 50 mM NH$_4$HCO$_3$ (pH 8.0). Beads were incubated with 1 $\mu$g Trypsin/Lys-C in 50 $\mu$l 50 mM NH$_4$OH and incubated at 37°C for 4 h in a thermomixer at 800 rpm. Next, the digest was separated from the beads, an extra 0.5 $\mu$g Trypsin/Lys-C was added and the digest was further incubated overnight at 37°C. Finally, the digest was centrifuged at 20,800 rcf in an Eppendorf cen-trifuge for 5 min, the supernatant was transferred to a new 1.5 ml Eppendorf tube, and the peptides were dried in a Speedvac and stored at −20°C until mass spectrometry (MS) analysis. Co-purified proteins were identified by mass spectrometry using an Orbitrap Elite (Thermo Fisher Scientific) or Q Exactive mass spectrometer (Thermo Fisher Scientific) using the procedures as described below. Proteins with at least two matched high confident peptides in at least two experiments in the dataset were retained. Background proteins were filtered out based on frequency of occurrence of the co-purified proteins in a large dataset containing 543 TAP experiments using 115 different baits (Van Leene et al, 2015). True interactors that might have been filtered out because of their presence in the list of nonspecific proteins were retained by means of semi-quantitative analysis using the average normalized spectral abundance factors of the identified proteins (Van Leene et al, 2015).

## Ultimate 3000 RSLC nano–Orbitrap Elite system (analysis of TAP data)

The obtained peptide mixtures were introduced into an LC–MS/MS system, the Ultimate 3000 RSLC nano (Dionex) in-line connected to an Orbitrap Elite Hybrid Ion Trap-Orbitrap Mass Spectrometer

(Thermo Fisher Scientific). The sample mixture was loaded on a trapping column (made in-house, 100 $\mu$m internal diameter [I.D.] × 20 mm [length], 5 $\mu$m C18 Reprosil-HD beads, Dr. Maisch GmbH). After back-flushing from the trapping column, the sample was loaded on a reverse-phase column (made in-house, 75 mm I.D. × 150 mm, 5 $\mu$m C18 Reprosil-HD beads, Dr. Maisch). Peptides were loaded with solvent A (0.1% trifluoroacetic acid and 2% acetonitrile) and separated with a 30-min linear gradient from 98% solvent A' (0.1% formic acid) to 50% solvent B' (0.1% formic acid and 80% acetonitrile) at a flow rate of 300 nl/min, followed by a wash step reaching 100% solvent B'. The mass spectrometer was operated in data-dependent, positive ionization mode, automatically switching between MS and MS/MS acquisition for the 20 most abundant peaks in a given MS spectrum. In the Orbitrap Elite, full scan MS spectra were acquired in the Orbitrap at a target value of $3 \times 10^6$ with a resolution of 60,000. The 20 most intense ions were then isolated for fragmentation in the linear ion trap, with a dynamic exclusion of 20 s. Peptides were fragmented after filling the ion trap at a target value of $3 \times 10^4$ ion counts.

### Ultimate 3000 RSLCnano–Q exactive (analysis of TAP data)

The obtained peptide mixtures were introduced into an LC–MS/MS system, the Ultimate 3000 RSLC nano (Dionex) in-line connected to a Q Exactive Mass Spectrometer (Thermo Fisher Scientific). The sample mixture was loaded on a trapping column (made in-house, 100 $\mu$m I.D. × 20 mm (length), 5 $\mu$m C18 Reprosil-HD beads, Dr. Maisch GmbH). After back-flushing from the trapping column, the sample was loaded on a reverse-phase column (made in-house, 75 mm I.D. × 150 mm, 5 $\mu$m C18 Reprosil-HD beads, Dr. Maisch). Peptides were loaded with solvent A (0.1% trifluoro-acetic acid and 2% acetonitrile) and separated with a 30-min linear gradient from 98% solvent A' (0.1% formic acid) to 50% solvent B' (0.1% formic acid and 80% acetonitrile) at a flow rate of 300 nl/min, followed by a wash step reaching 100% solvent B'. The mass spectrometer was operated in data-dependent, positive ionization mode, automatically switching between MS and MS/MS acquisition for the five most abundant peaks in a given MS spectrum. The source voltage was 3.6 kV and the capillary temperature was 275° C. One MS1 scan (m/z 400–2,000, AGC target $3 \times 10^6$ ions, maximum ion injection time 80 ms), acquired at a resolution of 70,000 (at 200 m/z), was followed by up to five tandem MS scans (resolution 17,500 at 200 m/z) of the most intense ions fulfilling predefined selection criteria (AGC target $5 \times 10^4$ ions, maximum ion injection time 80 ms, isolation window 2 D, fixed first mass 140 m/z, spectrum data type: centroid, intensity threshold $1.3 \times 10^4$, exclusion of unassigned, 1, 5–8, >8 positively charged precursors, peptide match preferred, exclude isotopes on, dynamic exclusion time 12 s). The higher energy collisional dissociation (HCD) collision energy was set to 25% Normalized Collision Energy and the polydimethylcyclosiloxane background ion at 445.120025 D was used for internal calibration (lock mass).

### Fluorescent protein-immunoprecipitation (FP-IP)

Sterilized seeds were germinated on $\frac{1}{2}$ Murashige and Skoog (MS) medium. For standard conditions, plants were grown for 7 d under continuous light at 22°C. For cisplatin treatment, seedlings were grown on $\frac{1}{2}$ MS medium for 6 d (6 dag). Then, they were transferred to $\frac{1}{2}$ MS plates supplemented either with or without 50 $\mu$M cisplatin and were grown for another day. For the different growth regimes and

treatments used in the meta-analysis, see Table S4, sheet1. 150–200 seedlings were harvested, frozen in liquid nitrogen and ground with a TissueLyser (QIAGEN) (30 Hz, 4 × 30 s). Total proteins were extracted as described in Henriques et al (2010). Total protein extracts (4 mg/IP) were immunopurified using anti-GFP antibody coupled to 50 nm size magnetic beads (MACS Technology, Miltenyi) with a method from Hubner et al (2010) and Horvath et al (2017) and digested in column with trypsin (Promega). The resulting peptide mixture was desalted before LC–MS/MS analysis (Omix C18 100 $\mu$l tips; Varian) and the purified peptide mixture was analyzed by LC–MS/MS using a nanoflow RP-HPLC (Lc program: linear gradient of 3–40% B in 100 min, solvent A: 0.1% formic acid in water, solvent B: 0.1% formic acid in acetonitrile) on-line coupled to a linear ion trap-Orbitrap (Orbitrap-Elite or Fusion-Lumos; Thermo Fisher Scientific) mass spectrometer operating in positive ion mode. Data acquisition was carried out in a data-dependent fashion, the 20 most abundant, multiply charged ions were selected from each MS survey for MS/MS analysis (MS and HCD spectra were acquired in the Orbitrap, and collision-induced dissociation spectra in the linear ion trap).

### Data analysis FP-IP (Fig S2)

To identify potential bait-specific interactors, MaxQuant proteomics software version 1.6.6.0 (Cox & Mann, 2008) was used to perform label-free quantification analysis on the corresponding MS files (.raw) and *A. thaliana* database. Recommended default parameters were used with the minimum ratio count set to 1. Given the label-free approach, multiplicity was also set to 1. False discovery rate threshold for peptide and protein identification was kept at 1%. The resulting non-normalised data were extracted for downstream analysis in R statistical software environment (version 4.0.0: "Arbor Day") interacting with RStudio v1.2.5033 (RStudio).

Quantified proteins were first segregated by bait and filtered for identification by site, matched to the reverse decoy database or common laboratory contaminants. The filtered list of proteins was then further trimmed to allow for a maximum of up to two missing values in each bait experimental condition. The raw intensities for the remaining proteins were normalized using variance-stabilizing normalization approach, and the missing values statistically imputed by randomly drawing from a left shifted Gaussian distribution (shift = 1.8; width = 0.2) via the DEP package (Zhang et al, 2018). As part of quality control, all samples were assessed for technical artefacts and outliers using principal component analysis and agglomerative hierarchical clustering performed on the top 500 most variable proteins. Hierarchical clustering was performed using complete linkage method with Euclidian distance as a proximity measure. In addition, Pearson's correlation coefficients were calculated to further access the reproducibility across samples.

Differential protein expression analysis was performed using limma as part of the DEP package. To correct for multiple hypothesis testing, *P*-values were adjusted by the Benjamini–Hochberg procedure. For each bait, significant interactors were determined based on appropriate logarithmic FC and adjusted *P*-value thresholds. Results were visualised via volcano plots using R.

### Data analysis FP-IP (meta-analysis and cisplatin treatment)

Raw data were converted into peak lists using the in-house Proteome Discoverer (v 1.4) (Guan et al, 2011) and searched against the

Swissprot database (downloaded 2019/6/12, 560,292 proteins) using the Protein Prospector search engine (v5.15.1) with the following parameters: enzyme: trypsin with maximum one missed cleavage; mass accuracies: 5 ppm for precursor ions and 0.6 D for fragment ions (both monoisotopic); fixed modification: carbamidomethylation of Cys residues; variable modifications: acetylation of protein N-termini; Met oxidation; cyclization of N-terminal Gln residues allowing maximum two variable modifications per peptide. Acceptance criteria: minimum scores: 22 and 15; maximum E values: 0.01 and 0.05 for protein and peptide identifications, respectively. Another database search was also performed using the same search and acceptance parameters except that Uniprot.random.concat database (downloaded 2019/6/12) was searched with *A. thaliana* species restriction (89,229 proteins) including additional proteins identified from the previous Swissprot search (protein score > 50).

FCs of the proteins upon cisplatin treatment were determined by using the Proteome Discoverer (v 2.4.1.1) (Thermo Fisher Scientific) software using MS1 quantitation.

For the meta-analysis, 182 IPs (50 GFP, 35 E2FA, 39 E2FB, 20 E2FC, 32 RBR1, 3 DPA and 3 DPB) were used. The statistical analyses were performed by edgeR (Robinson et al, 2010) using spectral counting (Branson & Freitas, 2016) to determine relative abundance of individual proteins (label-free quantitation). As cut-offs, we used a *P*-value of 0.05 and a FC of eight relative to the negative controls.

### Plant materials

The *A. thaliana* accession Columbia-0 (Col-0) was used as the wild type reference. All mutants and transgenic lines used in this study were in the Col-0 background. The mutant lines *aly1-1* (SALK_073108), *aly1-2* (SALK_114476), *aly1-3* (SAIL_409_B01), *aly2-2* (SALK_118765C), *aly2-3* (GK_083C03), *aly2-4* (SALK_056946), *aly3-1* (SALK_40756), *aly3-3* (SALK_49711), *aly3-4* (SALK_125138C), *lin37a-2* (SALK_057175), *lin37b-3* (SALK_103139C), *lin52b-1* (GK_854A11), *tcx5-1* (SALK_047165), *tcx5-2* (SALK_144605C), and *tcx6-1* (GK_453H07) were identified from the GABI-KAT, SAIL, and SALK T-DNA collections (Sessions et al, 2002; Alonso et al, 2003; Kleinboelting et al, 2012) and provided by the Nottingham Arabidopsis Stock Centre (NASC) (Scholl et al, 2000). The mutant line *nac044-1* was kindly provided by Prof. Masaaki Umeda (Nara Institute of Science and Technology) and was described previously (Takahashi et al, 2019). It was used as the genetic background for the *PRO_{NAC044}:mEGFP:NAC044* line as described below. The *lin52a-c1* mutant was generated by CRISPR/Cas9 (Fauser et al, 2014) using the protospacer-containing oligonucleotides listed in Table S5. For the FP-IP experiments, we used Col-0 plants expressing translational GFP- or CFP-fusions, that is, *E2FA:GFP* (Magyar et al, 2012), *E2FB:GFP* (Kállai et al, 2020), *E2FC:GFP* (Kállai et al, 2020), *DPA:GFP* (see below), and *DPB:3xCFP* (see below) under the control of their own promoter, as well as *PRO_{35S}:GFP* (Magyar et al, 2012) as control. Construction of the *PRO_{TCX5}:TCX5:EGFP*, *PRO_{LIN37A}:LIN37A:EGFP*, and *PRO_{LIN37B}:LIN37B:EGFP* reporter lines is described below. For confocal analysis, plants expressing *pgE2FA-3xvYFP* and *pgE2FB-3xvYFP* were used (Leviczky et al, 2019; Őszi et al, 2020).

### Plasmid construction and plant transformation

For generation of the *PRO_{NAC044}:mEGFP:NAC044* reporter, a 3,936-bp genomic sequence of *NAC044* was amplified by PCR and subsequently integrated into the *pENTR2B* vector by SLiCE reaction (Zhang et al, 2014). After introducing a *SmaI* restriction site in front

of the start codon, the obtained construct was sequenced and an *mEGFP* fragment was introduced into the *SmaI* site. For generating *PRO_{TCX5}:TCX5:EGFP*, *PRO_{LIN37A}:LIN37A:EGFP*, and *PRO_{LIN37B}:LIN37B:EGFP* reporter constructs, genomic fragments of *TCX5* (5,650 bp), *LIN37A* (4,381 bp), and *LIN37B* (3,862 bp) were amplified by PCR and cloned into *pDONR201* vector by Gateway BP reaction. The resulting plasmids were sequenced and used for creating C-terminal EGFP fusions, by inserting *EGFP* fragments in frame at the position corresponding to the C-terminus of the protein encoded by each gene. All fusion constructs were then transferred into the binary destination vector *pGWB501* (Nakagawa et al, 2007) by Gateway LR reaction. To construct the *PRO_{DPA}:DPA:GFP* and *PRO_{DPB}:DPB:3xCFP* translational fusions, the promoter regions and the genomic clones including exons and introns were amplified from genomic DNA (Col-0) using the primer combinations described in Table S5. The coding sequence of a single GFP or a triple CFP was added as a C-terminal fusion to the genomic sequence of *DPA* and *DPB*, respectively, in the *pGreenII*-based *pGII0125* destination vector (Galinha et al, 2007) by using the Invitrogen 3 way gateway system (Invitrogen). Transgenic plants were generated by *Agrobacterium*-mediated transformation (Zhang et al, 2006).

To generate constructs for yeast two-hybrid assays, the coding sequences of the respective genes were amplified from cDNA and attB-recombination sites were added in two consecutive PCRs. By Gateway BP reactions, these sequences were subcloned into the *pDONR223* entry vector. The corresponding N-terminally fused *pGAD424-GW*, *pGADT7-GW*, *pGBKT7-GW*, and *pGBT9-GW* destination clones as well as the C-terminally fused *pGADCg* and *pGBKCg* destination clones were generated by Gateway LR reactions. Primers used for construct generation are shown in Table S5.

### Root growth assay

Plants were germinated and grown on vertical plates containing $\frac{1}{2}$ MS medium under long day conditions (16 h light, 8 h dark) at 22°C for 5 d. Seedlings were then transferred to $\frac{1}{2}$ MS medium containing cisplatin (Sigma-Aldrich) or MMC (Sigma-Aldrich) in the indicated concentrations and were grown for further 6 or 7 d. It is to note that the optimal cisplatin concentration required a new adjustment for every new batch of cisplatin, being in our hands around 10–15 μM. After 5 d, the position of the root tip was marked. Plates were scanned and root length was measured digitally using the Simple Neurite Tracer plugin (Longair et al, 2011) for ImageJ.

### Yeast two-hybrid assay

Yeast two-hybrid assays were performed according to the Yeastmaker Yeast Transformation System 2 manual (Clontech). The yeast strain AH109 was co-transformed with an AD-fused and a BD-fused construct using the lithium acetate method. Yeast cells harbouring both constructs were grown on DDO, TDO, and QDO medium (−L/−W, −L/−W/−H, and −L/−W/−H/−Ade, respectively) to assess protein/protein interactions. Co-transformation of a construct with the corresponding mEGFP construct was used as an auto-activation control.

### Microscopy

Plants were germinated and grown on vertical plates containing $\frac{1}{2}$ MS medium under continuous light at 22°C for 5 d. Seedlings were

then transferred to $\frac{1}{2}$ MS medium containing cisplatin (Sigma-Aldrich) or MMC (Sigma-Aldrich) in the indicated concentrations and were imaged at different time points for time course experiments. For this, roots were placed in 0.1 mg ml$^{-1}$ propidium iodide (PI) solution and fluorescence was imaged by confocal laser scanning microscopy using an LSM780 (Zeiss) with a 40× water immersion C-Apochromat 1.2 NA objective (Zeiss). The microscope was controlled using the Zen black software (Zeiss). GFP variants and PI were exited with a 488-nm argon laser and a 561-nm DPSS laser, respectively. GFP variant fluorescence was detected at 498–550 nm and PI fluorescence was detected at 568–690 nm.

### Expression analysis

Total RNA was extracted from 7-d-old Arabidopsis seedlings using the innuPREP Plant RNA kit (Analytik Jena BioSolutions) according to the instructions of the manufacturer. cDNA synthesis was performed using a QuantiTect Reverse Transcription Kit (QIAGEN) following the manufacturer's instructions. The cDNA was used for semi-quantitative PCR experiments to test for the presence of mRNA in respective T-DNA insertion lines. Primers used for semi-quantitative PCR experiments are listed in Table S5.

### EdU labelling of arabidopsis root tip

To detect S phase cells, a commercially available 5-ethynyl-2′-deoxyuridine (EdU) kit was used (Click-iTTM EdU Alexa FluorTM 488 Imaging Kit; Thermo Fisher Scientific). Stocks of EdU and reaction mixture components were prepared in accordance with manufacturer's instructions. Seedlings were incubated in 5 $\mu$M EdU-containing liquid MS medium for 30 min.

After the incubation period, seedlings were placed on a glass microscope slide containing a drop of 3.7% vol/vol formaldehyde (Sigma-Aldrich) and shoots were excised. The cut roots were transferred to a 1.5 ml microcentrifuge tube containing 1 ml of fixation solution consisting of 3.7% vol/vol formaldehyde + 0.1% vol/vol Triton X-100 (Sigma-Aldrich) in microtubule-stabilizing buffer (MTSB) for 1 h under vacuum at RT. The fixation solution was removed, and the roots were washed three times with MTSB. Samples were permeabilized in 1 ml 0.5% vol/vol Triton X-100 in PBS for 15 min at RT. Permeabilization solution was discarded and roots were washed three times with PBS. Samples were then incubated in Click-iT reaction mixture for 40 min at RT, protected from light. Samples were washed three times with PBS. For counter-staining of the nuclei, samples were incubated in 500 $\mu$l 25% vol/vol Sysmex CyStain UV Precise P staining buffer in PBS for 15 min at RT, protected from light. Samples were then washed three times with PBS. Samples were kept in PBS after the final wash until imaging.

### In silico analysis

Proteins sequence alignments were carried out using the MUSCLE algorithm from the European Molecular Biology Laboratory-European Bioinformatics Institute toolkit with default settings (Madeira et al, 2019). Alignments were formatted at http://www.bioinformatics.org/sms/multi_align.html setting the coloring option to 50%. Gene expression analysis of publicly available mRNA seq data from wild-type

Arabidopsis samples was performed using the GENEVESTIGATOR software (Hruz et al, 2008).

## Data Availability

Supporting data for Fig 1 (TAP results) can be found in Tables S1 and S2. Enrichment data for the GST-pulldown experiments, graphically represented by the volcano plots in Fig S2, are listed in Table S3. Fig 2 presents a reduced dataset of the GST-pulldown meta-analysis detailed in Table S4.

## Supplementary Information

## Acknowledgements

We thank the Single Cell Omics Advanced Core Facility staff of the Hungarian Centre of Excellence for Molecular Medicine (HCEMM) and Biological Research Center for help with their resources and their support. HCEMM has received funding from the EU's Horizon 2020 research and innovation program (739593). This work was supported through a fellowship of the University of Hamburg to L Lang, a grant by the Development and Innovation Office of Hungary (GINOP-2.3.2-15-2016-00032) to A Pettkó-Szandtner, a grant from the Japan Society for the Promotion of Science KAKENHI (20H05408 and 18H04833) to M Ito, a grant from the Hungarian National Research Funding (NKFIH-K132486) to Z Magyar, a BBSRC-NSF grant (BB/M025047/1) to L Bögre, and a DFG grant (SCHN 736/16-1) to A Schnittger.

### Author Contributions

L Lang: formal analysis, investigation, and visualization.
A Pettkó-Szandtner: conceptualization, formal analysis, investigation, and visualization.
H Tunçay Elbaşi: formal analysis, investigation, and visualization.
H Takatsuka: formal analysis, investigation, and visualization.
Y Nomoto: formal analysis, investigation, and visualization.
A Zaki: formal analysis, investigation, and visualization.
S Dorokhov: formal analysis, investigation, and visualization.
G De Jaeger: resources, data curation, supervision, funding acquisition, and project administration.
D Eeckhout: data curation, formal analysis, and investigation.
M Ito: resources, supervision, funding acquisition, project administration, and writing—review and editing.
Z Magyar: conceptualization, formal analysis, supervision, funding acquisition, investigation, visualization, and writing—review and editing.
L Bogre: conceptualization, supervision, funding acquisition, writing—original draft, and project administration.
M Heese: conceptualization, formal analysis, supervision, visualization, writing—original draft, and project administration.
A Schnittger: conceptualization, formal analysis, supervision, funding acquisition, writing—original draft, and project administration.

## Life Science Alliance

**Conflict of Interest Statement**

The authors declare that they have no conflict of interest.

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
