## [Reviewer comments · Life Science Alliance]

Life Science Alliance

The DREAM complex represses growth in response to DNA damage in Arabidopsis

Lucas Lang, Aladar Pettkó-Szandtner, Hasibe Elbasi, Hirotomo Takatsuka, Yuji Nomoto, Ahmad Zaki, Stefan Dorokhov, Geert De Jaeger, Dominique Eeckhout, Masaki Ito, Zoltan Magyar, Laszlo Bogre, Maren Heese, and Arp Schnittger

DOI: <https://doi.org/10.26508/lsa.202101141>

Corresponding author(s): Arp Schnittger, Universität Hamburg and Maren Heese

Review Timeline:

Submission Date:	2021-06-23
Editorial Decision:	2021-06-23
Revision Received:	2021-09-13
Editorial Decision:	2021-09-14
Revision Received:	2021-09-17
Accepted:	2021-09-17

Transaction Report:

Please note that the manuscript was previously reviewed at another journal and the reports were taken into account in the decision-making process at Life Science Alliance.

June 23, 2021

Re: Life Science Alliance manuscript #LSA-2021-01141-T

Prof. Arp Schnittger
University of Hamburg
Department of Developmental Biology
Biozentrum Klein Flottbek
Ohnhorststr. 18
Hamburg 22609
Germany

Dear Dr. Schnittger,

Thank you for submitting your manuscript entitled "The DREAM complex represses growth in response to DNA damage in Arabidopsis" to Life Science Alliance. Please upload the manuscript, revised according to the Reviewer comments. Please also include a point-by-point response to those comments. I expect to be able to handle this quickly once I receive these files.

I include here my initial response when I offered transfer:

Eric Sawey, the Executive Editor of Life Science Alliance, would be interested in publishing this work if it is revised according to the Reviewer comments. Reviewer 2's 2nd point regarding the mechanism through which DREAM sub-units are required for growth arrest in response to DNA damage will not be necessary for acceptance at LSA.

Thank you for this interesting contribution to Life Science Alliance. We are looking forward to receiving your revised manuscript.

Sincerely,

Eric Sawey, PhD

- A letter addressing the reviewers' comments point by point.
- An editable version of the final text (.DOC or .DOCX) is needed for copyediting (no PDFs).
- High-resolution figure, supplementary figure and video files uploaded as individual files: See our detailed guidelines for preparing your production-ready images, <https://www.life-science-alliance.org/authors>
- Summary blurb (enter in submission system): A short text summarizing in a single sentence the study (max. 200 characters including spaces). This text is used in conjunction with the titles of papers, hence should be informative and complementary to the title and running title. It should describe the context and significance of the findings for a general readership; it should be written in the present tense and refer to the work in the third person. Author names should not be mentioned.

B. MANUSCRIPT ORGANIZATION AND FORMATTING:

Detailed response to the reviewer comments on Lang and Pettkó-Szandtner et al. (LSA-2021-01141-T)

Our responses to the reviewer comments are in blue font.

Referee #1:

The authors employ variety of *in vivo* experiments, following standard protocols, designed to fish out potential members of known transcriptional regulatory complexes related to cell cycle control. These investigations begin with a TAP tag data set from a suspension culture upregulated for DDR response (thus broadening the data from previous studies in uninduced suspension cultures). Most significantly, the authors identify two LIN52 homologs, an important component of the DREAM complex in animals which was not previously identified in plants. Although identification of a plant homolog of an animal gene is not usually big news, this was a missing piece of a complex that is highly conserved in animals, and not previously identified either *in silico* or through pull down studies in non-induced cultures. Thus the plant DREAM complex is very like the animal complex. Some novel potential interactors with components of the MuvB core of the DREAM complex were also identified. The requirement for the LxCxE motif for binding by RBR1 was tested for several of the proteins identified as RBR1 interactors in this study, and the authors find that while the proteins in the plant and animal complexes are homologous, the mechanism of interaction is slightly different.

Because protein-protein contact assays are highly prone to experimental artifact (both false negatives and false positives), the authors go on to test these interactions using a variety of established *in vivo* systems (in plants and yeast) for discovering/testing protein-protein contacts. Some contacts are confirmed, others are not (but remain potential interactors) and additional potential contacts identified.

To better understand the function of these components in plants, they generate mutations in the endogenous candidate cell cycle control genes and phenotype the resulting plants for root growth and response of root growth to *cis-plat*. They demonstrate that the single mutants are still competent for growth arrest in response to damage, but it's not clear whether there are minor effects on either growth or arrest. The obvious next step is to test double or triple KOs, as several of these genes have closely related homologs in *Arabidopsis*, and so there may easily be functional redundancy. They do demonstrate an important role for the LIM37 homologs (only recently identified in *Arabidopsis*, in another study) as redundant contributors to the inhibition of growth in response to *cis-plat*, using the double mutant.

Unfortunately they do not investigate the effects of double or triple mutants in other duplicated genes (the ALY (Lin9 homolog) 1,2,3), or, especially of interest for this study, LIN52A,B.

We thank this reviewer for his/her time to review our manuscript and the constructive feedback. We are also happy to see the positive judgment of our work as a very useful resource holding many interesting observations (summary statement of this reviewer).

We completely agree, that the proposed multiple mutant analysis is a logical and interesting next step. However, we are facing another competitive situation concerning the DREAM complex as apparently another paper is currently under review reporting on the role of the DREAM during plant stress response. For this reason, we decided to focus on the functional analysis of selected components. In our first manuscript version, we focused on LIN37, as a representative component of the MuvB-core complex, since we saw interaction for LIN37 and NAC044 in the Y2H. And as representatives for non-core components, we chose to analyze the conventional E2F transcription factors. To address this reviewer's comments, at least partially, we have now in addition generated all *aly* double mutant combinations and studied their root growth in this revised manuscript (Fig S8D; line 404 ff). However, we do not see an enhanced growth phenotype in comparison to the wildtype when treated with MMC. Thus, it seems likely the redundancy expands to the level of the triple mutant. However, this triple mutant has recently been shown to be lethal (Ning et al. 2020, DOI: 10.1038/s41477-020-0710-7). Therefore, to explore this point further, one needs more delicate genetic engineering, e.g. inducible RNAi constructs, etc. We think that this goes beyond the level of our current manuscript and is also not the central point of our work in which we present complex composition and reveal a role of the DREAM complex in repressing growth during stress conditions.

The authors then provide phenotypic data on growth and DDR (cis-plat) (growth, cell cycle response, and PCD) in the root tip for the E2FB mutants. Many of these responses, to other damaging agents, have been previously described for E2FA and E2FC, but not E2FB.

The effects are subtle, there seems to be slightly less inhibition of growth in the *ef2b* mutants, which is a surprising given this protein's role as an activator, rather than suppressor, of cell proliferation. The authors don't note or discuss this (superficially) paradoxical observation.

In the plant field, the categorization of E2Fs into activator and repressor types is largely based on overexpression studies and therefore somewhat controversial and much work remains to be done with single and multiple mutant combinations. Moreover, it appears to be a general theme that many if not all transcription factors have both positive and negative roles in terms of transcriptional regulation. With that respect, we kindly refer to the work of some of the authors of this manuscript which have recently shown a repressive role of E2FB when associated with RBR during leaf development (Özi et al. 2020, DOI: 10.1104/pp.19.00212). Thus, the here-presented finding that the *e2fb* mutant is less sensitive to DNA damaging agents is in accordance with the paper by Özi et al. None-the-less, the reviewer comment made us aware that we should better discuss these findings and we have now included a paragraph on the positive and negative role of E2FB in our revised manuscript (line 596 ff).

The authors repeatedly claim that E2FB is required for PCD in root tips in response to Cis-plat. However, the mutant appears to be indistinguishable from wt in both the images and the data on "cell death area" (Fig 9CF). So this is a very misleading statement that should be removed from Results and Discussion.

There is a claim in the text that cell death in, specifically, columella cells (a cell type in the root cap), is decreased in the e2fb-2 mutant, but the corresponding figure describes this as lateral root cap/columella (in other words, the entire root cap, not just the columella).

Given that the root cap sloughs off during root growth, and cell death is observed there in the absence of exogenous DNA damaging agents, these root-cap specific claims need to be better supported by data with and without cis-plat. Either clarify or remove these studies- certainly remove the statement that this gene is required for PCD.

We thank the reviewer for this comment making us aware of the fact that we need to present these data in a clearer way. It has been previously recognized that the proliferating vascular cells respond differently to DNA damage in terms of cell death than the columella initials (columella plus the lateral root cap stem cells and their immediate daughters). We quantitated these cells as described before (Horvath et al 2017, DOI: 10.15252/embj.201694561). We also redid the statistical tests using ANOVA followed by Tukey post-hoc test as requested by reviewer 3 and we clearly find significant differences with respect to the columella/lateral root cap initial cells.

In summary,

This paper is a compilation of possibly DREAM-related observations, including a large number of proteomics studies (not really my field, so I might under- or over-appreciate these) with different mutants generated but subjected to different assays, and no assay of the double/triple mutants for the probably redundant gene function (just LIN37A,B).

So this paper lacks focus but still has many useful observations, particularly the discovery of the LIN52 homologs.

We thank one more time this reviewer for his/her constructive comments that have helped to improve our manuscript.

Smaller issues:

Editor or author: To make the reviewer's life a little easier, please include line numbers or at least page numbers. Also putting figure legends on the same page as figures is very helpful.

We apologize for lacking line numbers in the original manuscript. We added them in this revised version. For the figures, we followed LSA guidelines, which demand the figure legends to be in the text.

And note that I'm completely unqualified to judge the TAP procedure and related statistics.

Beginning of paper:

"...a complement of the entire DREAM complex known from...animals" What? Do they mean a component? Or do they mean "the entire complement of proteins in the DREAM complex"- yes, when I get to the Discussion I see they mean the later. Please use the suggested phrase in both the beginning and end of the paper.

Sentences using “complement” have been adjusted to be clearer.

On Page of "Results"

Other issues:

Figure 1: when two baits are shown connected by edges in figure 1, I can't tell if which bait captured the interaction described, or if both did. For example, TCX5 has a solid line to RBR1. I guess that describes using TCX5 as bait (because the authors didn't use both c and n terminal tags for RBR1). Was that interaction also captured, at all, using RBR1 as bait? Please make this clear- somehow- throughout. I realize the data is also in sup table 1. Maybe arrowheads could indicate connection from bait to prey, or all RBR1 bait-based observations (edges) could be red, or something. It would be nice to see which interactions are confirmed both ways. This is just a suggestion.

Thanks for pointing this out. We added this additional information (Fig 1).

The concentration(s) of cis-plat employed is only described for the seedling experiments. What dose was used for the liquid cell cultures and what is the effect of this dose on growth rate?

This was omitted by mistake. The concentration was 30 μ M, as we only did a 16 hrs treatment. We added the information to the TAP material and methods section (line 620) .

We usually use less (about 10 – 15 μ M) in long term root growth assays, since we want to avoid a complete growth arrest of the wildtype, to be able to see potential hypersensitivity as well as resistance in mutants analyzed. In our hands wildtype seedlings stop growing on 30 μ M cisplatin after about 1 day.

Figure 6A:

"When we analyzed root tips of the reporter line grown on control plates occasionally we saw strong nuclear accumulation of mEGFP-NAC044 in isolated cells in different tissue layers of the root (Figure 6A). Next, we "

Note figure 6A doesn't include control plate- maybe you just mean the zero time point? Let the reader know.

Yes, correct, in this case we need to refer to the zero time point - this has been rephrased (line 357-361). We also see this on plates without cisplatin, but since we did not include these in the time course experiments, time point zero is indeed the correct reference here.

P15? "On a whole seedling level..." I would point out to the reader that perhaps the majority of cells in a six day old seedling may be postmitotic, making signals harder to detect and quantify. If you agree.

By explicitly pointing out that we are looking at whole seedling level, we wanted to remind the reader, that we don't have cellular/tissue resolution and different cells might (and are even likely to) respond differently, depending on tissue context and cell cycle status. The reviewer raised an important point here since of course all effects that take place in mitotic cells will be diluted by large number of post mitotic cells in seedlings. We added some explanation to the text (line 433-439).

Figure 8, Methods for FP-IP? I think the authors have pasted in a method from a completely different study which includes agents and growth conditions not employed here. Ie, Inducing agent AZD-8055, light changes?, Nutrient level comparisons..

These conditions were actually used for the experiments analyzed in the meta-analysis. However, since we never looked at these experiments in detail here, we decided to delete this information from the general material and methods section because it is obviously confusing. However, the information can still be found in Table S4, which presents the experiments used for the Meta-Analysis.

P15 "since the lin37 mutants consistently showed better growth" I'm not seeing that, I guess you mean "better growth on cis-plat".

Yes, this was not precise. We rephrased (line 398 ff)

And all of these apparently redundant genes should be investigated as double or triple (for ALY).

The aly double mutants have been added (Fig S8D; line 404 ff), the triple is reported to be lethal (Ning et al. 2020, DOI: 10.1038/s41477-020-0710-7). For additional explanation see above, first response-paragraph.

P17: I disagree with the authors interpretation of their EdU -incorporation and DAPI staining and I think the authors might want to reconsider. Here's how I see it:

A decrease in the frequency of wild-type mitotic cells in response to cis-plat does not suggest induction of a "mitotic checkpoint" (which would trap cells in M phase and increase their frequency).

Calling it mitotic checkpoint was misleading. We were thinking about a G2/M transition checkpoint, not the SAC.

It suggests that cells are not progressing into M. This programmed inhibition of the cell cycle could be occurring at any stage of the cell cycle (other than M).

In contrast, the reduction in S phase cells in wild-type could be interpreted as arrest anywhere from G1 through S, since cells arrested intra-S (experiencing

lesion-stalled replication forks and programmed inhibition of origin firing) would be expected to exhibit less EdU incorporation and so probably will not be scored as S phase cells.

For the EdU and mitotic index experiment, we have used a short 3h cisplatin treatment and a 30min EdU pulse during this period. Considering that the total cell cycle length in root tips of Arabidopsis is ~18h, the DNA damage should have been initiated close to the events (S phase or mitosis) observed. However, we agree that this can only be seen as a first attempt to understand what is happening at a cellular level, so we decided to phrase this more carefully and rewrote this part of the E2FB section (line 455 ff, line 484 ff).

Referee #2:

In their manuscript entitled "The DREAM complex represses growth in response to DNA damage in Arabidopsis", Lang, Pettkó-Szandtner and colleagues describe a very nice biochemistry work providing robust identification of the plant DREAM complex in Arabidopsis. They clarify the composition of the plant DREAM complexes in cultured cell lines after DNA damage, and compare their data to IP experiments performed on untreated plantlets. Globally these results represent a very large amount of work, and provide nice and clear evidence (i) for the composition of prevalent DREAM complexes in plant cells and (ii) for the specific association of E2Fb and E2Fc, but not E2Fa to these complexes.

Authors go on by investigating the role of several DREAM sub-units in the response to DNA damage through reverse genetics approaches. This part clearly argues for the involvement of at least some components in the growth arrest triggered by DNA damage, although the focus on this specific role is maybe not fully justified, for example in the case of LIN37 (see below), and the depth of this analysis could be improved.

In spite of the quality of the work and of the writing, I think some key points are missing for this study to reach the novelty and provide the mechanistical insight required for publication in a generalist journal such as this one.

We also like to thank this reviewer for his/her comments and the appreciation of our work. Due to a situation of scientific competition on the DREAM subject, we decided to publish our work on the composition of the DREAM complex and its link to DNA damage control without going into great mechanistic details at this point. We accept that this scope might be too limited for th is Journal and therefore accept the proposition for transfer to LSA, which we think is a good fit.

1. The TAP and IP experiments combined to the meta-analysis summarized on Figure 2 are very nice and provide a very convincing view of what the composition of plant DREAM complexes is, although as authors point themselves, most sub-units were shown to interact before. The novelty here therefore rather lies into the systematic analysis and robustness of the data, rather than in the identification of previously unknown protein complexes.

We would like to point out, that in the so far most comprehensive DREAM complex paper of Ning et al. (doi: 10.1038/s41477-020-0710-7), the authors state the following:

“We did not test pairwise interactions for all DREAM components by Y2H. Although we detected the interaction of TCX6 with LIN37B, RBR1 and ALY3, we did not determine whether the TCX6 homologue TCX5 can interact with the three DREAM components (Supplementary Fig. 6). Similarly, we detected the interaction of ALY3 and LIN37B with other DREAM components by Y2H but did not determine whether the ALY3 and LIN37B homologues (ALY1, ALY2 and LIN37A) can interact with the other DREAM components (Supplementary Fig. 6). Therefore, more detailed interactions of DREAM components remain to be studied in future.”

Our Y2H matrix now fills this information gap.

Importantly, I think a control with untreated cells is missing to really decide how much of what authors observe has anything to do with DNA damage. Indeed, TAP experiments on cell cultures are performed exclusively in the presence of cisplatin, and are not even compared to previous TAP experiments performed with RBR in untreated cells (Van Leene et al 2010). I suppose this is probably because the quality of data obtained in this way has increased so much that the comparison would not be valid.

Yes indeed. We previously discussed with our colleagues from the TAP facility and technical improvement (e.g. a more sensitive detection system) precludes a 1:1 comparison of the TAP results, although the same expression construct was used. In the previous untreated RBR1 TAP, none of the MuvB-core components were identified and CDKB1;1 but not CDKA1;1 was co-precipitated with RBR1. Since the difference in the results might be of technical nature, we do not want to bring up this comparison in the paper.

Regardless, this makes it extremely difficult to figure out whether these interactions are promoted by DNA damage, or happen during "normal" cell cycle progression. Indeed, the comparison with GFP-IP performed in plantlets reveals very few differences, so in the absence of a control experiment performed in cell lines without cisplatin, authors cannot conclude that the composition of DREAM complexes they describe relates to DNA damage.

Yes, like others (Kobayashi et al. 2015, DOI: 10.15252/embj.201490899; Ning et al., DOI: 10.1038/s41477-020-0710-7), we see that plant DREAM complexes are also present in the absence of externally inflicted DNA damage. However, to investigate if there is additional complex formation upon damage, we performed a quantitative analysis of E2FB-GFP containing complex that can be precipitated from DNA-stressed and unstressed seedlings. Here, we see a up to 2 fold enrichment of E2FB-GFP and associated DREAM components after cisplatin treatment on a whole seedling level. Of course at this point we cannot say if this is the result of a mild elevation in all cells or a stronger change in a certain sub-population of cells.

To complement this observation, we now include in this revised paper version confocal pictures of E2FB-3xYFP expressing seedlings after 24 h growth on cisplatin containing

plates compared to control plates. This already indicates globally enhanced accumulation of E2FB after treatment with genotoxin (Figure 8A).

For example, on p6, they write "However, none of our affinity purifications contained the activating transcription factor MYB3R4, suggesting the prevalence of repressive DREAM complexes after DNA damage" but MYB3R4 is not found in co-IPs performed in untreated plantlets, so it may just be that repressive DREAM complexes are more prevalent whatever the conditions.

Due to the large number of dividing cells in cell culture, we would expect here rather activating than repressing DREAM complexes, which is not the case for seedlings, that include many differentiated cells, which have exited from the cell cycle. So for the sake of this argument the two situations cannot be compared. However, we accept that our statement might be too speculative at this point, so we decided to delete it.

2. Based on results shown on Figures 7, 9, S8 and S9, it seems that some DREAM sub-units are required for growth arrest in response to DNA damage. However, authors do not at all investigate how this happens. They do not follow the expression of DDR marker genes in wild-type and mutants with and without damage, or ask the question of DREAM binding to these putative targets. The specific association of NAC044 to DREAMs in response to DNA damage would clearly suggest that it may target the whole complex to specific genes, but this is not at all explored.

These are definitely questions that need to be addressed in the future and we accept that reviewer sees this as a requirement for publication at another Journal level. However, we feel that our current dataset is already very extensive and provides sufficient new insights to justify a publication in LSA.

Other points

3. In Figures 3, 4 and 5, authors summarize Y2H results and test the functionality of the LxCxE motif found in NAC044, TCX5 and TCX6 for binding to RBR. The results are clear but would still need to be confirmed in planta through co-IP.

Yes, this would be the next step, but we feel beyond the scope of this paper.

4. Authors focus on the role of LIN37A and B in DNA damage response, but *lin37a* mutants are smaller than the wild-type in control conditions, so authors could just as well emphasize the role of LIN37A as a growth promoting factor in the absence of DNA damage.

Since we already have evidence that there are activating and repressing versions of the DREAM complex (Kobayashi et al. 2015, DOI: 10.15252/embj.201490899), we are not surprised to see this phenotype. However, also the *lin37a* linked shorter root growth under control conditions was not significant in all of our experiments (albeit repeatedly

seen) so we do not want to put too much weight on this observation. To not give a misleading impression, we explicitly added to the text that this reduced growth was only seen occasionally (line 403-404).

5. On Figure 8, authors show that E2FB containing DREAM complexes are enriched after DNA damage. Can they see an increased accumulation of E2FB after DNA damage?

In this revised version we provide now a confocal images of E2FB-3xvYFP and E2FA-3xvYFP expression in roots after cisplatin treatment, which indicates an accumulation of E2FB under genotoxic stress compared to control conditions. Also it shows that not all tissues are affected to the same degree (Figure 8A, line 418 ff).

Minor points

Abstract : Upon persistent damage, plant growth and cell proliferation ARE reduced.

We have corrected this mistake.

In supplemental Table 1, DRC1 is highlighted in orange because it shows homology to a DREAM subunit in animals whereas DRC2 isn't. I suppose this means that DRC2 is a plant-specific component, authors may want to state this explicitly in the text?

Yes –we added this statement (line 245).

p9-10 and at the top of p18 references are in italics

We have corrected this.

On Figure 9, authors quantify S-phase cells in root tips of wild-type and ef2b mutants in the presence or absence of cisplatin. Rather than counting and absolute number S-phase cells, it would make more sense to show a proportion of labelled cells in the meristem, to avoid effects that could result from changes in the total number of cells in meristems.

We assumed that during the short 3h treatment the meristem size is unlikely to change, but in principle we agree that expressing S-phase and mitosis in percentage of all the cells are more correct. Therefore, we counted all the DAPI-stained nuclei and calculated the mitotic index and S-phase index for this revised version (Fig 9D, Fig S9B)

Also, statistical analysis of the data shown in panels G and H would better be done through ANOVA followed by Tukey post-hoc test. This would allow determining

whether the apparent increase in the number of S-phase cells in the e2fb1 mutant (and to a lesser extent in the e2fb2 mutant) compared to Col0 on MS medium is statistically relevant.

We redid the statistical analysis using ANOVA + Tukey and got a statistically significant difference only for EdU but not for mitosis. In the confocal section, as the reviewer pointed out there are very few mitotic cells that can be considered. Therefore, this result is only a trend rather than a significant result and we reworded accordingly. (line 455 ff, line 484 ff, Fig 9D, Fig S9B)
Also please note, that under control conditions, the relative S-phase count is not higher in the e2fb mutants than in Col-0 (Fig 9D)

Supplemental Figure 9A is missing statistical analysis of the data.

We added the statistics to the time course of the root growth assay (Fig S9A).

I am surprised at the small number of mitosis per root tip. I am wondering whether this could be due to difficulties in capturing mitosis in deep cell layers of the meristem, and whether it would not be better to squash roots for more accurate counting.

We only counted the cells in one central optical section, however for 20-30 roots per assay. Squash preparation would improve capturing more mitotic cells and observe mitotic abnormalities, but would make the determination of meristem border problematic. Thus, we would like to follow our current experimental procedure.

I don't think the quality of images shown on Supplemental Figure 9C really allows recognizing "abnormal chromosome alignment", assessing mitotic defects would require more thorough counting, probably on squashes or at least with a higher magnification to be able to identify lagging chromosomes etc... This is not central to the manuscript and could thus also simply be removed.

We agree and removed this part.

Referee #3:

The DREAM complex plays very important roles in cell cycle regulation. In this manuscript, the authors characterized the DREAM complex in Arabidopsis through affinity purification followed by mass spectrometry analysis. They identified all homologues of animal DREAM complex as well as some new components including NAC044 and SOG1. They further demonstrated that E2FB and LIN37 were required for growth inhibition upon DNA damage. The authors did a lot of work and provided some new information about the plant DREAM complex and their involvement in DNA damage responses.

We thank also reviewer 3 for his/her time and energy helping us to improve our manuscript. The feedback was very helpful and we are also happy about the general appreciation of our work.

1. All core components of the Arabidopsis DREAM complex except LIN52 have been identified previously (Kobayashi et al. 2015; Ning et al. 2020). This makes this study complementary.

Yes, this work was done in parallel and we only learned from the Ning et al work when it become published. None-the-less, our work also holds many new findings since Ning et al. have focused on the role of the DREAM in regulating DNA methylation and not DNA damage. For instance, we provide a full Y2H interaction matrix and find evidence for a modified MuvB-core/RBR1 interaction interface when compared to the human complex.

2. The title needs to be modified. The authors claim that the DREAM complex represses growth in response to DNA damage. However, the components of DREAM complex have different effects on plant growth. For example, it is well-known that RBR and E2Fs function antagonistically. After DNA damage treatment, the amiRBR lines had more dead cells than WT (Horvath et al. 2017), while the e2fb mutant had fewer dead cells.

We think, with the chosen title, we do not claim that the only function of the DREAM complex is to repress growth after DNA damage, but we state that this is one of it's functions. It is clear by several publications, that individual DREAM components can function in different assemblies and exert other than growth repressive functions (Derkacheva et al. 2013, Kobayashi et al. 2015, Ning et al. 2020). However, if the general notion is that our title implies an exclusive growth repressive function, we are happy to change this. Maybe this can be discussed with the editor.

3. The authors mentioned the "direct" protein interaction in several places. For example, "NAC044 directly binds to RBR1 and to the MuvB-core component LIN37" in Discussion. Neither IP nor Y2H data can determine whether the interactions are direct or not. In vitro pull-down assays using purified proteins may be an option for this purpose. It will be very informative to show which protein interactions are direct.

In our opinion, the Y2H system is used to test for direct interaction between two proteins, the bait and the prey. Of course it cannot be excluded that (related or unrelated) yeast proteins mediate interaction between bait and prey – but this would be then considered as a false positive result.

We think it is common knowledge that false positives (as well as false negatives) occur in the Y2H system and that results have to be taken with a grain of salt. None-the-less, Y2H is a powerful interaction assay system and it is a commonly used. We hope the reviewer agrees that this assay is a valid first step to test for direct interaction. The next level could indeed be to perform in vitro interaction assays (however, these assays, as the reviewer knows, also have their drawbacks, e.g. due to mis-folding and missing posttranslational modifications if proteins are expressed in bacteria).

4. Since LIN52 was not previously identified, it will be more interesting to test the role of LIN52 in DNA damage responses than E2Fb and LIN37.

We agree, that it is very interesting to investigate LIN52 as well. However, we do not think it is more interesting to look at LIN52 than LIN37 when it comes to the role of DREAM in DNA damage. We focused here on LIN37 as representative of the MuvB-core because LIN37, according to our Y2H results, most likely directly interacts with NAC044. E2FB on the other hand was chosen as a representative of the non-core components.

5. In Figure 6, what are the images in the right corner of each panel? Please describe in the figure legend.

These are the PI stainings indicating which part of the root is shown in the confocal picture. We added this information to the legend and thank the reviewer for spotting this omission (line 955 ff).

6. For Figure 7B, 9B, 9G, and 9H, to test the difference between Col-0 and mutants after treatment, please use two-way ANOVA instead of Student's t-test.

We redid the statistics using ANOVA and the appropriate post-hoc tests (Figs 7B, 9B, 9D, S9B).

7. For Figure 9D, please mark the mitotic cells by arrows.

We now moved the statistical analysis of the mitotic index to the supplement where we show a DAPI-only picture including arrows pointing at the mitotic cells (Fig S9B-C).

September 14, 2021

RE: Life Science Alliance Manuscript #LSA-2021-01141-TR

Prof. Arp Schnittger
Universität Hamburg
Department of Developmental Biology
Institute of Plant Sciences and Microbiology
Ohnhorststr. 18
Hamburg 22609
Germany

Dear Dr. Schnittger,

Thank you for submitting your revised manuscript entitled "The DREAM complex represses growth in response to DNA damage in Arabidopsis". We would be happy to publish your paper in Life Science Alliance pending final revisions necessary to meet our formatting guidelines.

- please add ORCID ID for the corresponding author-you should have received instructions on how to do so
- please add your main, supplementary figure, and table legends to the main manuscript text after the references section
- please revise the legend for Figure S2 so it matches with the figure
- please add callouts for Figures 3A, B; 7A, B; 8D; S2A-D; S5A, B; S6A-D; S7A-D; S8A-C; S9E, F to your main manuscript text
- if possible, please try to present the figure on one page
- In the Data Availability statement, please specify which supporting data are available in which supplemental sections

A. FINAL FILES:

B. MANUSCRIPT ORGANIZATION AND FORMATTING:

Sincerely,

September 17, 2021

RE: Life Science Alliance Manuscript #LSA-2021-01141-TRR

Prof. Arp Schnittger
Universität Hamburg
Department of Developmental Biology
Institute of Plant Sciences and Microbiology
Ohnhorststr. 18
Hamburg 22609
Germany

Dear Dr. Schnittger,

Thank you for submitting your Research Article entitled "The DREAM complex represses growth in response to DNA damage in Arabidopsis". It is a pleasure to let you know that your manuscript is now accepted for publication in Life Science Alliance. Congratulations on this interesting work.

DISTRIBUTION OF MATERIALS:

Again, congratulations on a very nice paper. I hope you found the review process to be constructive and are pleased with how the manuscript was handled editorially. We look forward to future exciting submissions from your lab.

Sincerely,
